# Habitual intake of fat and sugar is associated with poorer memory and greater impulsivity in humans

**Martin R. Yeomans**[1]*, **Rhiannon Armitage**[1], **Rebecca Atkinson**[2], **Heather Francis**[3], **Richard J. Stevenson**[3]

**1** School of Psychology, University of Sussex, Brighton, United Kingdom, **2** Centre for Dementia Studies, Brighton and Sussex Medical School, Brighton, United Kingdom, **3** Department of Psychology, Macquarie University, Sydney, New South Wales, Australia

* martin@sussex.ac.uk

**Data Availability Statement:** The data along with analysis scripts and outputs area publicly available from the University of Sussex research repository (https://doi.org/10.25377/sussex.22068731).

## Abstract

The vicious cycle model of obesity suggests that repeated habitual intake of a diet high in fat and sugar (HFS) results in impairment in hippocampal function which in turn increases impulsive behaviours, making it harder to resist unhealthy diet choices. Evidence from studies with rodents consistently show switching to a HFS diet impairs performance on hippocampally-sensitive memory tasks. The limited literature in humans also suggest impaired memory and increased impulsivity related to higher habitual HFS intake. However, these changes in memory and impulsivity have been looked at independently. To investigate how these effects are inter-related, three experiments were conducted where relative HFS intake was related to measures of memory and impulsivity. In Experiment 1 (90 female participants), HFS was associated with higher scores on the Everyday Memory Questionnaire-revised (EMQ), and higher scores on the total, Attention (BISatt) and Motor (BISmot) sub-scales of the Barratt Impulsiveness Scale (BIS11). Experiment 2 (84 women and 35 men), replicated the association between HFS and EMQ, and also found HFS related to poorer performance on the hippocampally-sensitive 4 mountain (4MT) memory task. The association between HFS intake and the BISatt replicated, but there were no significant associations with other BIS11 measures or delay-discounting for monetary rewards. Experiment 3 (199 women and 87 men) replicated the associations between DFS and 4MT and EMQ, and also found an association with overall recall, but not response inhibition, from a Remembering Causes Forgetting task: HFS was also significantly associated with BIS total, BISatt and BISmot. In all three studies these associations remained when potential confounds (BMI, age, gender, hunger state, restrained and disinhibited eating) were controlled for. Mediation analysis found that the effect of HFS on memory at least part mediated the relationship between HFS and impulsivity in Experiments 1 and 3, but not 2. Overall these data provide some support for the vicious cycle model, but also suggest that trait impulsivity may be a risk factor for poor dietary choice.

**Funding:** he author(s) received no specific funding for this work.

**Competing interests:** The authors have declared that no competing interests exist.

## Introduction

It has long been recognised that poor nutrition, and in particular deficits in key essential nutrients, can negatively impact cognitive function (see [1,2]). What is perhaps more surprising is that recent studies suggest that overconsumption of foods can also have negative effects on some aspects of cognition, particularly memory (see [3–5] for recent reviews). In particular, there is now abundant evidence from studies in rodents that consuming a diet high in fat or fat and sugar (HFS) can lead to a specific deficit in memory tasks that place high demands on the hippocampus (e.g. [6–8]). What is surprising is that these deficits develop rapidly: evidence of impaired memory has been reported within a week of rats being switched to a HFS diet [9]). Alongside these studies, there have now been a smaller number of studies with human volunteers that also suggest that habitual intake of diets high in fat and sugar is associated with poorer memory [10,11]. There is also some evidence that higher intake of fat and sugar is associated with higher impulsivity in humans [12]. However, how these effects on memory and impulsivity are inter-related remains unclear, and therefore the aims of the experiments in the present paper were to test further the inter-relationships between habitual HFS intake, memory and impulsivity, and in particular to test whether these represent separate or related effects.

One of the notable features in animal studies of effects of HFS diets on memory is the specificity of these effects (e.g. [9,13]). Thus, rats placed on a HFS diet showed a deficit in performance on a spatial memory task but performance on object recognition was unimpaired. As spatial memory is known to involve the hippocampus [14], these data have been interpreted as evidence for a specific effect of HFS diets on hippocampal function, supported by evidence from animal studies of altered hippocampal structure and function after consuming HFS diets (reviewed by [15,16]). Data also suggest parallel effects in human studies. Most studies have examined how individual variation in habitual intake of HFS relates to performance on memory tasks. Notably, higher habitual HFS intake was associated with poorer performance on two tasks known to involve the hippocampus (verbal paired associates and the logical memory subtests from the Wechsler Memory Scale), but not on tasks which have lower hippocampal involvement (the Trail Making task and Wisconsin Card sorting task: [11]). Subsequent studies replicated the association between higher HFS and performance on the Wechsler Memory Scale [17], and a verbal paired associates task [10]. A weakness in this approach is that the data are correlational, but there have also been two short intervention studies where volunteers overconsumed diets with high fat and sugar content for short periods. In the first [18], four days consuming a breakfast high in fat and sugar resulted in poorer memory retention on the Hopkins-Verbal Learning Test, relative to a control group, and this effect was then replicated in a larger sample of participants [19]. Thus, the limited human literature to date is consistent with the idea that habitual fat and/or sugar intake is associated with poorer memory performance, particularly on tasks that involve the hippocampus, and the two intervention studies suggest these effects can emerge after relatively short exposures, in line with the studies in rats.

If habitual HFS intake does impair memory, it might then be expected that individuals with higher HFS in their diet may be aware of these memory deficits. To test that idea, here we explored whether habitual HFS intake was associated with higher scores on a validated questionnaire measure of memory lapses, the Everyday Memory Questionnaire revised (EMQ: [20]). If habitual higher intake of a diet high in fat and sugar does truly impair memory, then it might reasonably be expected that people would have noticed instances of forgetting in their normal lives, and so would score higher on the EMQ measure.

There is also some evidence that higher habitual HFS intake is associated with higher levels of impulsivity [12]. In that paper, an initial study found that higher HFS intake was associated

with higher overall scores on the short form [21] of the widely used Barratt Impulsiveness Scale (BIS: [22]). Further analysis using the three second-order BIS factors found the association with HFS to be evident with the BIS motor and non-planning sub-scales, but not the attention subscale. In their Study 2, a subset of participants from the first study completed a wider battery of impulsivity measures, with evidence that higher HFS intake was associated with more Urgency measured using the Urgency, Premeditation, Perseverance and Sensation Seeking scale (UPPS: [23]), and in performance on a food- but not money-based delay-discounting task. We are not aware of any replications of these findings, and the second aim of the present studies was to try and extend the literature on the association between HFS intake and impulsivity.

To date, the associations between HFS intake, memory and impulsivity have been reported separately. However, theoretically it is important to determine whether these apparently separate findings represent two different effects of habitual diet or are different expressions at a cognitive level of one effect. With impulsivity, we contrast two alternative explanations why greater impulsivity may be associated with higher habitual intake of HFS diets (Fig 1). The first of these posits that those with higher trait impulsivity may be more prone to unhealthy diet choices, and so may be more attracted to the immediate sensory rewards that fat and sugar may provide (the trait model: Fig 1A). This idea is consistent with studies that have reported associations between measures of impulsivity and obesity (see [24,25] for reviews), and between impulsivity and traits known to predict overconsumption such as the disinhibition scale from the Three Factor Eating Questionnaire [26–28]. Those and other findings suggest that impulsivity is a risk factor for overeating and consequent obesity, with increased reward reactivity as the likely mediator [29]. However, the majority of the literature relating impulsivity and obesity is correlational, and has not included measures of habitual diet, focussing instead either on measures of body-size or food-cue reactivity. Therefore, whether impulsivity is a cause or consequence of overconsumption cannot readily be determined from that literature.

The second explanation suggests that the association between impulsivity and HFS intake may be a consequence of the effects of HFS diets on memory (the vicious cycle model, VCM: Fig 1B). This idea is made implicitly in the vicious cycle model of obesity [30,32]. That model suggests that the effects on the brain of overconsumption of HFS diets directly causes the reported memory impairment and, since the brain areas involved in that effect (particularly the hippocampus) are also involved in regulating reactions to reward stimuli such as food cues [33,34], and food cue reactivity is itself related to more general impulsivity [31], this in turn leads to greater reward-sensitivity [35] and consequent increased impulsive decision making. The consequent decrease in ability to resist the immediate rewards of consuming a HFS diet then exacerbates the effect, leading to a vicious cycle that then leads to obesity. Crucially the VCM does not specify that the hippocampal dysfunction caused by a Western style diet is specific to food/food memory and so the impairment can be measured using the type of general memory measures we employed in the three experiments reported below. Thus we test the wider implication of the VCM that it is this general diet-induced impairment to memory that alters response to food cues: when sated, food cues would not normally result in memory retrieval because of the role of the hippocampus in memory inhibition [36]. Consequently, non-specific inhibition of memory retrieval induced by overconsumption results in greater excitatory food memory retrieval when sated.

How then to dissociate these two ideas? According to the trait model, higher impulsivity is a risk factor for greater HFS intake, so the primary prediction of the trait model is that impulsivity will be associated with higher DFS scores. In contrast, the VCM (Fig 1B) explicitly suggests that the effects on memory result in increased impulsive decision making in response to

# A)Trait model

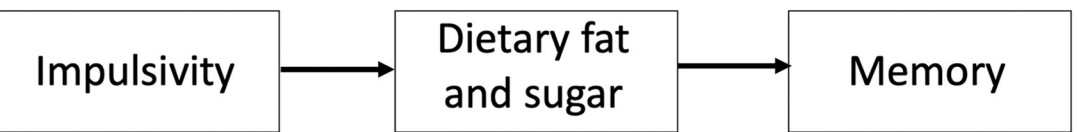

# B) Vicious cycle model

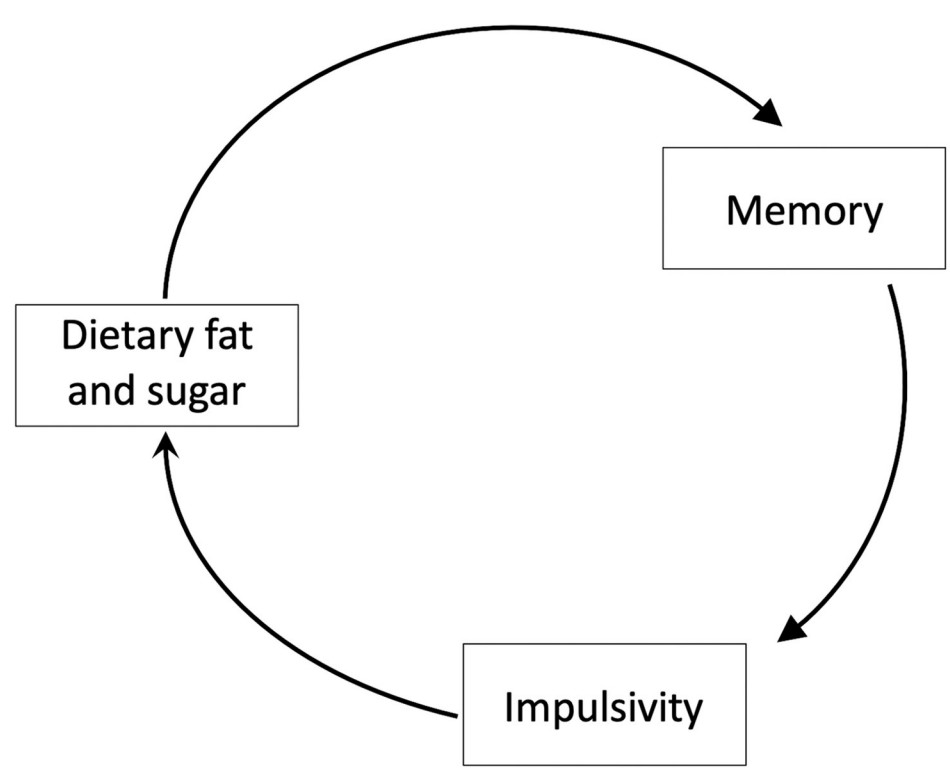

**Fig 1. Schematic representation of two alternative ways memory and impulsivity might be related to habitual intake of diets high in fat and sugar (HFS).** (A) trait impulsivity is associated with higher HFS intake, which in turn leads to impaired memory. (B) HFS intake leads to impaired memory and consequent increased reactivity to food cues [30], which is associated with increased impulsivity (e.g. [31]), and which further enhances HFS intake.

food, and therefore the association between HFS intake and impulsivity would be mediated by the changes in memory. By measuring aspects of memory and impulsivity alongside habitual HFS intake for the first time, we can contrast these two ideas. If the association between HFS intake and impulsivity is mediated by the changes in memory, that would be clear support for the VCM. What then might the trait model say about the relationship between diet and memory? Here we make a secondary prediction that trait impulsivity is associated with higher DFS scores, and that higher DFS scores then are associated with impaired memory: so any effect of impulsivity on memory is mediated by the impact of impulsivity on DFS scores.

To further complicate these inter-relationships, it is also known that individual differences in factors beyond diet are associated with memory and impulsivity. The first of these is body-size, with clear evidence for cognitive impairments associated with obesity [37–39]. Most relevant here are studies that relate relative body-size (as measured by the widely used body-mass index, BMI, metric: [40]) to specific measures of memory. For example, performance on a specific spatial episodic memory task decreased with increasing BMI [41]. To control for this, we firstly deliberately targeted a younger, healthier population where incidence of obesity is lower, and secondly measured BMI to control for this in analyses. Secondly, performance on memory tasks has been widely shown to vary with age, with cognitive decline evidenced widely in older populations [42]. We therefore also recorded age, again by targeting a younger healthy population but controlling for age in analyses. Finally, as noted above studies have also found evidence that impulsivity is related to known individual differences in eating behaviour, including cognitive restraint [43,44] and particularly opportunistic eating as measured by the Three Factor Eating Questionnaire (TFEQ) disinhibition scale [26,27,45]. We therefore included the TFEQ to test whether apparent associations between HFS, memory and impulsivity were indirect consequences of wider trait differences in eating attitudes. We assessed these inter-relationships in three separate experiments, each building on successive outcomes and limitations.

## Experiment 1

### Materials and methods

**Design.** The study examined the relationship between self-reported consumption of a high fat and sugar diet (measured by the Dietary Fat and Free Sugar Questionnaire, DFS), self-reported memory failure (measured by the EMQ) and impulsivity (measured by the BIS-11 and Dickman Impulsivity Inventory, DI), while controlling for eating attitudes (measured using the TFEQ), age and BMI.

**Participants.** Ninety female volunteers took part in the study. As no previous study had explored how habitual diet measured using the DFS related to memory measured using the EMQ, we could not conduct a reliable power-analysis, and based the sample size on an that estimated to detect a medium sized effect with power of 0.8. Potential participants were informed that the study involved a series of short online questionnaires exploring the relationship between eating habits and cognitive functioning. All participants gave their informed consent online before beginning the study. The study conformed to the British Psychological Society guidelines for ethical human research, and the study protocol was approved by the University of Sussex School of Psychology ethics committee (approval ER/LB369/1).

**Estimation of habitual fat and sugar intake.** The DFS [11,46], a validated food frequency questionnaire, provided a measure of habitual HFS intake. The DFS consists of 26 Likert- scale questions that measure intake of both saturated fat and added sugars. Participants were required to recall the number of times they consumed certain foods (such as dairy, red meat, cakes) or drinks over the preceding 12 months, with each item scored on a 1–5 scale, with 1 the least frequent and 5 the most frequent for each item. DFS scores can range from 26 to 130, and higher scores indicate a poorer quality diet that is higher in saturated fat and added sugar. Scoring generates both an overall total score as well as measures for frequency of use of items high in fat (DFS-fat), sugars (DFS-sugar), and both fat and sugar (DFS-fatsugar). The measure has established test-retest reliability and validity [46].

**Everyday memory measure.** The revised version of the EMQ (20) was developed as a validated shortened (13-item) version of the original 28-item Everyday Memory Questionnaire [47], with each item scored on a 5-point Likert scale ranged from once a month or less (0) to once or more a day (4). Royle and Lincoln [20] confirmed the reliability of the revised scale, and their analysis suggested the EMQ generated two factors which they interpreted as retrieval (7 items) and attentional tracking (4 items).

**Barratt Impulsiveness Scale (BIS 11; 22).** The BIS11 consists of 30 test items, scored using a 4-point Likert scale, ranging from rarely/never (1) through to almost always (4), measuring three dimensions of impulsivity: attention (BISatt), motor (BISmot) and non planning (BISnp). It was selected because of its wide use, high test-retest reliability and internal consistency [21,22,48], its wider association with trait eating and reactivity to food cues [26,49], and the previous finding that BIS-11 scores associated with dietary intake measured using the DFS [12].

**Dickman Impulsivity Inventory (DI, 50).** The DI was originally designed to measure functional (DIfunc, 11 items) and dysfunctional impulsivity (DIdys, 12 items), and the two subscales have been shown to have good reliability [50,51]. Subsequent research suggests that the dysfunctional scale is related to many other self-report measures of impulsiveness, including the BIS 11 [52], but the functional scale measures a more unique aspect of impulsivity and was included in this study for that reason.

**Three Factor Eating Questionnaire (TFEQ: [53]).** The TFEQ identifies individual differences in three dimensions of human eating behaviour: cognitive restraint (TFEQ-R, 21 items), disinhibited eating (TFEQ-D: 16 items), now often characterised as opportunistic eating [54] and a third factor targeting trait sensitivity to internal cues (hunger). In relation to the present study, the TFEQ-R and TFEQ-D scales were of relevance since past research has suggested that both restraint [55] and disinhibition [26] may be associated with individual differences in impulsivity which could have confounded the key focus on associations between the DFS measures and impulsivity.

**Procedure.** Data were collected in two parts: TFEQ data were collected between September 2015 and March 2016 as part of the recruitment questionnaire for a database of participants willing to participate in research in the Sussex Ingestive Behaviour Unit. Two hundred women who had completed that recruitment questionnaire were sent details of a second online study examining eating habits and cognition, and 90 of these women responded positively to that invitation: data collection was conducted between December 2015 and March 2016, with no more than two months between completion of the TFEQ and the online follow-up. The four questionnaires (DFS, EMQ, BIS11 and DI) were collated into a single online survey using Bristol Online Survey software, and all potential participants were sent a link to the survey along with a personal identifier which allowed their data to be linked to existing TFEQ data. The survey included confirmation of consent on the opening page, followed by the DFS, EMQ, BIS11 and DII in that order. After completion, participants were asked to record their current height, weight and age, and were given the opportunity to enter their email into a prize

draw with a single £25 reward. Self-report height and weight were used to estimate body mass index based on the standard formula ($kg/m^2$). Although self-reported BMI is less accurate than actual (weighed) BMI, large-scale studies have confirmed the acceptability of self-report BMI in cross-sectional and epidemiological studies (e.g. [56]).

**Data analysis.** Prior to analysis, data were checked for outliers (using visual inspection of box-plots and then analysis of z-scored data to formally define outliers) and any violations of normality. To test the first two objectives, we used step-wise regression to assess how firstly memory (EMQ scores) and secondly impulsiveness (BIS11 and DI scores) were related to overall habitual fat and sugar intake, and reported data are standardised beta coefficients. In step 1 we examined the relationship between memory/impulsivity and diet alone (e.g. EMQ ~ DFS), then in step 2 we included possible confounds of BMI and age (e.g. EMQ ~ DFS + BMI + age), and Step 3 TFEQ restraint and disinhibition scores (e.g. EMQ ~ DFS + BMI + age +TFEQD + TFEQR). ANOVA were used to test whether adding in the additional factors significantly increased the goodness of fit of each regression model. Where normality analysis raised concerns we used robust linear models using the lmRob model from the robust R package. Where an overall effect of diet was observed at Step 1, supplementary analyses examined whether these overall effects were also observed for the different sub-components of the EMQ or sub-scales of the impulsivity measures. Because of the large number of tests this entailed, significance was corrected for multiple comparisons using the Bonferroni adjustment. To aid interpretation of the relationships between impulsivity and diet measures, we also examined how the different impulsivity measures were inter-correlated using (Pearson correlations: see Supplemental data S1 Table).

To contrast the two models of how impulsivity, DFS and memory scores were inter-related, we conducted two separate mediation analysis. If the vicious-cycle model (Fig 1B) is correct, then any effect of DFS scores on impulsivity would be mediated by memory, and that mediation was tested using the Lavaan package in R [57]. Note we also ran additional models including BMI, age and TFEQD as covariates, but adding in these covariates had minimal impact on the models and are not reported for brevity. Finally, we compared the goodness of fit of these models using the Lavaan ANOVA function to see which model accounted for the most variance in the data.

All analyses were conducted using RStudio, and both the dataset and R-analysis scripts are available at 10.25377/sussex.22068731.

## Results

**Participant characteristics.** Summary data and participant characteristics are shown in Table 1. One participant declined to provide height and weight data: there were no other missing data. Four participants were obese (BMI >30): three of were formally outliers, but there was minimal difference in BMI between the lowest outlier (31.22) and the fourth case (31.11). Likewise with age, most participants were young (aged 18–26), but six were aged >29, again skewing the data. We therefore ran robust models when including age and BMI as factors conducting the regression and mediation analyses.

**Table 1. Demographic characteristics of participants in Experiment 1 (mean, SD and range).**

|  | n | Mean | Min | Max | SD |
|---|---|---|---|---|---|
| Age (years) | 90 | 22.6 | 18 | 66 | 7.2 |
| BMI ($kg/m^2$) | 89 | 22.3 | 15.9 | 48.9 | 4.5 |
| TFEQ-R | 90 | 8.2 | 0 | 21 | 5.3 |
| TFEQ-D | 90 | 7.2 | 1 | 15 | 3.9 |

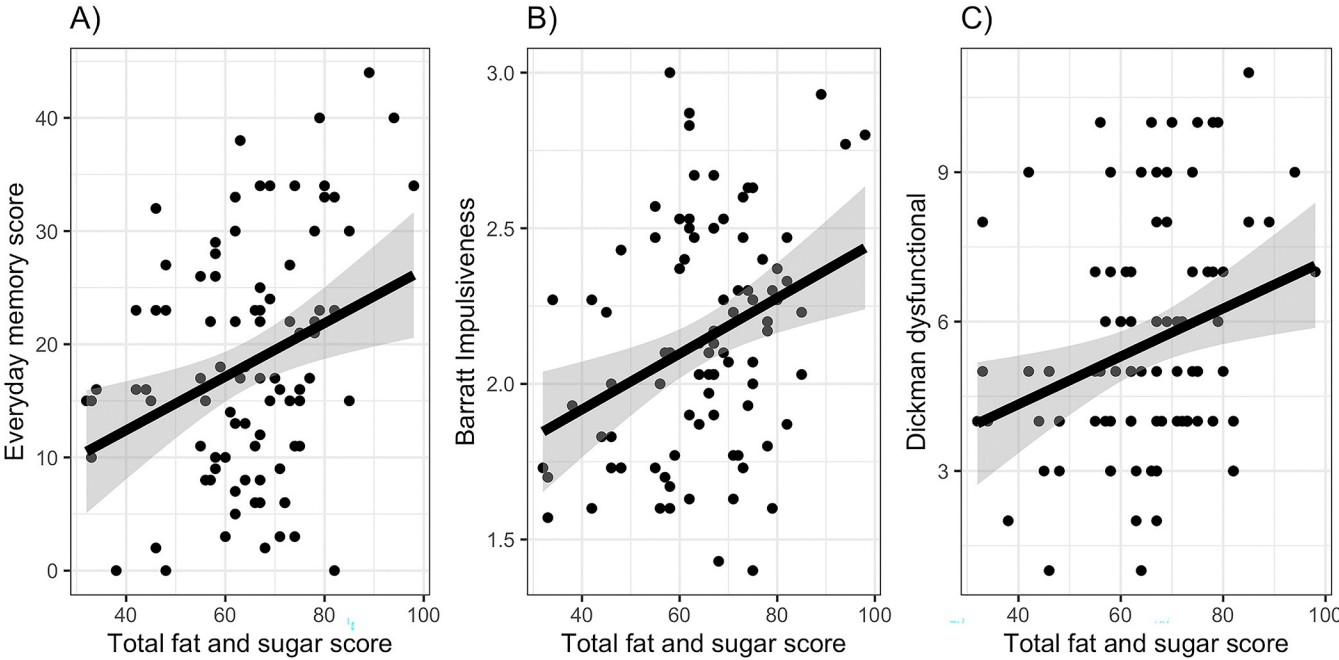

**Fig 2.** Memory and impulsivity measures related to overall fat and sugar intake (estimated using the DFS) from Experiment 1: (A) the Everyday Memory Questionnaire, (B) total score on the BIS11, (C) the dysfunctional scale from the DI.

**Diet and everyday memory.** There were no outliers in either the EMQ or DFS data. The overall model relating EMQ to HFS was significant ($r^2$ = 0.10, F(1,89) = 9.34, p = 0.003), with a positive association between EMQ and total DFS score (Fig 2A: $b$ = 0.24 [0.08, 0.39]). Step 2 confirmed this effect ($b$ = 0.28 [0.12, 0.43], p = 0.001) when controlling for age and BMI: neither age or BMI were significant predictors of EMQ scores. Likewise, the relationship between DFS and EMQ was still evident when participants with obesity were excluded ($b$ = 0.28 [0.11–0.44], p < 0.001). Model 3, including the two TFEQ factors, did not improve overall fit, and neither TFEQ factors predicted EMQ score.

Follow-up regressions then tested whether the relationship between EMQ and DFS was stronger for either of the two sub-scales of the EMQ (summarised in Table 2, row 1). As inclusion of BMI, age or TFEQ factors did not improve the model fit in the main analysis we only

**Table 2. Associations between the EMQ subscales and habitual intake of fat and sugar measured by the DFS in Experiments one and two.**

| Experiment | Everyday Memory Questionnaire scores | |
| --- | --- | --- |
| | Retrieval | Attentional tracking |
| One | 0.29** (0.09, 0.50) | 0.28** (0.07, 0.48) |
| Two | 0.39*** (0.22, 0.55) | 0.37*** (0.20, 0.54) |

Data are unstandardised beta scores (95% confidence intervals) from regression models.

(*** p<0.001

** p<0.01

* p<0.05, adjusted within each experiment for multiple comparisons).

**Table 3. Associations between subscales of the BIS and habitual intake of fat and sugar measured by the DFS in the three Experiments.**

| Experiment | Barratt impulsiveness subscale | | |
| --- | --- | --- | --- |
| | Attention | Motor | Non-planning |
| One | 0.37 ** | 0.20 | 0.25 * |
| | (0.17, 0.57) | (-0.01, 0.41) | (0.05, 0.46) |
| Two | 0.25 ** | 0.10 | 0.07 |
| | (0.07, 0.43) | (-0.08, 0.28) | (-0.11, 0.26) |
| Three | 0.19 ** | 0.16* | 0.12 |
| | (0.07, 0.30) | (0.05, 0.28) | (0.01, 0.24) |

Data are unstandardised beta scores (and 95% confidence intervals) from linear regression models.

(** p<0.01

* p<0.05, adjusted for multiple comparisons within each Experiment).

looked at simple effects here. Table 2 provides little evidence that the relationship between EMQ and DFS scores was specific to a sub-component of EMQ.

**Diet and impulsivity.** Overall BIS11 score was related to the total DFS score (Fig 2B: F (1,89) = 10.60, p = 0.002: $b$ = 0.009 [0.003, 0.014]). Adding in age and BMI confirmed the effect of DFS ($b$ = 0.36 [0.16, 0.56]), and also found BIS11 scores decreasing with increasing BMI ($b$ = -0.21, [-0.4, 0.00]). Repeating this analysis with participants with obesity excluded confirmed the overall association between DFS and BIS11 total ($b$ = 0.33 [0.12, 0.53], p = 0.002), but now found no significant effect of BMI. Adding in TFEQ scores confirmed the relationship between BIS and DFS scores but found no associations between BIS11 and either TFEQ scores.

We then explored the relationships between the three BIS11 sub-scales and DFS total score, using the full dataset (Table 3, row 1): there were significant correlations between DFS and the attention and non-planning BIS11 sub-scales, but not motor subscale.

The model relating DIfunc and total DFS score was not significant (F(1,88) = 1.74, p = 0.19: $b$ = 0.14, [-0.07, 0.35]), but DIdys was associated with higher DFS total scores (Fig 2C: F(1,88) = 7.54, p = 0.007: $b$ = 0.28 [0.08, 0.48]), and this effect remained significant when controlling for BMI and excluding participants with obesity.

**Mediation analysis.** Since both EMQ and some measures of impulsivity were related to DFS scores, we then tested whether EMQ mediated the relationship between DFS and impulsivity, as implied by the vicious cycle model (Fig 1B). First we focussed on the BIS total score, and checked whether BIS II and EMQ scores were related, and a Pearson correlation confirmed that they were (r(88) = 0.47, p = < .001, 95% CI = [0.29, 0.62]). Mediation analysis confirmed that without EMQ there was a significant direct relationship between DFS and BIS11 (b = 0.009, 95% CI [0, 0.01], t = 3.72, p = <0.001), but including EMQ in the model increased the size of that direct effect (b = 0.015, 95% CI [0.01, 0.02], t = 4.36, p = <0.001), with a significant indirect effect (b = 0.003, 95% CI [0, 0.01], t = 2.51, p = 0.012), as can be seen in Fig 3A. Thus there was some evidence that EMQ mediated the relationship between BIS 11 and DFS scores, in line with the vicious cycle model.

In contrast, the trait model (Fig 1A) would suggest that higher scores on the BIS 11 would increase DFS, and consequently also increase EMQ. We therefore also tested this using mediation (Fig 3B). This confirmed that there was a significant relationship between BIS11 and EMQ (b = 13.081, 95% CI [8.38, 17.79], t = 5.45, p = <0.001), and that was still evident when DFS was included (b = 11.478, 95% CI [6.82, 16.13], t = 4.83, p = <0.001). However, the indirect effect between BIS11 and EMQ was not significant (b = 1.603, 95% CI [-0.41, 3.62], t = 1.56, p = 0.119).

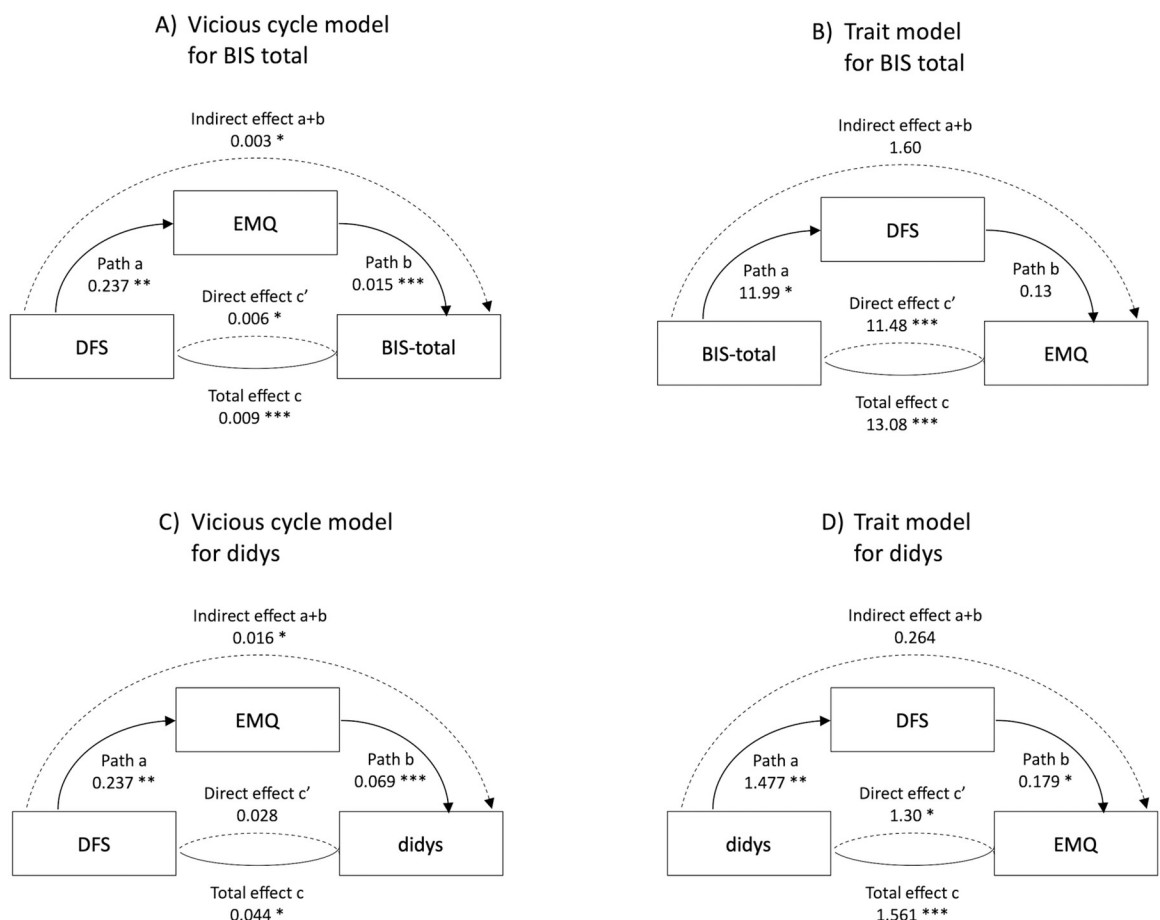

**Fig 3. Summary diagrams of the four mediation models relating fat and sugar intake measured using the DSF, memory (Everyday Memory Questionnaire, EMQ or Four Mountains task, 4MT) and impulsivity (Barratt Impulsiveness Inventory, BIS11 or Dickman Dysfunctional Impulsivity, DIdys).** The pathways are labelled with the standard mediation coefficient labels, with the b-values for each relationship. * p< 0.05, ** p<0.01, *** p<0.001.

Having constructed these two models, we then intended to contrast the goodness of fit of these models. However, in both cases the final solution from the mediation analysis generated models with 0 degrees of freedom, results which confirm that both models were very well fitted to the observed data [58], but making it impossible to conduct any test comparing the goodness of fit.

As the DIdys measure was also associated with higher DFS scores, we repeated these two mediation models using the DIdys measure. With the vicious cycle model (Fig 3C), the indirect pathway where the effect of DFS on DIdys was mediated by EMQ was significant (b = 0.016, 95% CI [0.001, 0.031], t = 3.144, p = 0.032). In contrast, with the trait model, although all three direct pathways were significant (see Fig 3D and S2 Table for details), the indirect pathway linking DIdys to EMQ via DFS was not significant (b = 0.264, 95% CI [-0.027, 0.556], t = 1.78, p = 0.076).

**Inter-relationships between impulsivity measures.** In this sample, the three BIS sub-scales were all significantly inter-correlated, and likewise the two Dickman scales were also significantly correlated. The only significant correlation between BIS and Dickman measures was between the BIS attention sub-scale DIdys factor (see S1 Table).

## Discussion

Experiment 1 found evidence that participant's own awareness of everyday memory failures, measured by the EMQ, and some aspects of impulsiveness (the total and sub-scale BIS scores and DIdys) were all related to habitual dietary intake of fat and sugar as estimated by the DFS. These relationships remained significant when participant age, BMI and eating attitudes (TFEQ-R and TFEQ-D) were controlled for. Overall, these data add to the existing literature relating habitual fat and sugar intake to poorer hippocampal memory performance [10,11] and greater impulsivity [12].

The second aim was to contrast two alternative models of how habitual diet, memory and impulsivity were inter-related. In relation to the VCM, there was evidence that the effect of DFS on EMQ significantly mediated the relationships between DFS and BIS11 and DFS and DIdys. Thus the suggestion that impaired memory because of habitual HFS intake in turn increased impulsivity was supported, but with a small effect size (every 1pt increase in DFS score increasing BIS11 by 0.003pts through its effect on EMQ, compared with 0.006pts directly). In relation to the trait model, again the mediation model fitted the data well but here there was no evidence that the relationship between BIS11 and DFS mediated the relationship between BIS11 and EMQ. However, the strong fit between the tested models and data in both cases meant we could not determine which model was the better description of these data.

Experiment 1 did however have a number of limitations. Firstly, it only included a self-report measure of memory impairment (the EMQ). While there is some evidence from clinical studies that greater memory impairment measured using the EMQ is associated with reduced hippocampal size [59], in Experiment 2 we included an objective measure of spatial memory, the Four Mountains task (4MT), which was developed by Hartley et al. [60] to target hippo-campal spatial function, and which has been shown to be a sensitive index of hippocampus spatial function [61,62]. We are unaware of any study that has focussed on the impact of habit-ual diet on spatial memory in humans, while noting that the animal literature has consistently found specific spatial deficits after HFS exposure (reviewed in 3). Secondly, we only looked at self-report measures of impulsivity in Experiment 1. Since previous research has suggested that performance on a behavioural measure of impulsivity, monetary delay discounting, may also relate to overweight, obesity and unhealthy diets (see [63] for review), we also included a widely used measure of delay discounting as well as the BIS 11 to explore further associations between habitual diet and impulsivity. Finally, Experiment 1 relied on self-report measures of height and weight, did not control for how hungry participants were at the time of testing, and tested a female-only sample. We included hunger since we are hypothesising that exposure to HFS diets impairs memory and through an effect on the hippocampus, and the experience of hunger has also been hypothesised to relate to hippocampal function [36]. We aimed to con-trol for these limitations in Experiment 2.

## Experiment 2

### Methods

**Design.**  The study examined the relationships between impulsivity (measured by the BIS11 and DDT) and objective memory (4MT) and self-reported memory failure (EMQ) with self-reported consumption of a high fat and sugar diet (measured by the DFS questionnaire), while controlling for hunger at the time of testing, eating attitudes (measured using the TFEQ), age and BMI.

**Participants.**  One hundred and 20 volunteers (84 Women and 36 men) took part in the study, recruited as in Experiment 1. Sample size was based on power-analysis of the

relationship between total DFS and EMQ scores from Experiment 1. To replicate that relationship with power = 0.8, the estimated sample size needed was 79. However, as we now added in sex as an additional factor, we increased sample size by 50%, to give n = 120. The study protocol was approved by the University of Sussex School of Psychology ethics committee (approval ER/MS804/1).

**Estimation of habitual fat and sugar intake.** Habitual diet was again assessed by the DFS, as in Experiment 1.

**Memory assessments.** We again used the EMQ to try and replicate Experiment 1, but also used the Four mountains task (4MT), which has been released as an App developed jointly by University College London and the University of York, and was administered on an iPad mini handheld tablet. Full details of the task can be found at: http://fourmountains.org.uk. The task presents 15 discrete trials: on each trial, a target image displaying four mountains in a particular topographical alignment was displayed for 10 sec, and after a short delay (2sec), the participant was presented (for 30 seconds) with four images, one of which was a rotated version of the target and the other three decoy (foil) images of similar four mountains. Participants had to select which image matched the earlier target. The number, out of 15, that were correctly identified provided the key memory score.

**Assessment of impulsivity.** We again used the BIS 11. Additionally, delay discounting was assessed using the 27-item choice Kirby Delay Discounting Task [64], although the monetary values were changed from US dollars to UK pounds to make this appropriate for the study population. The 27-item task presents choices which vary in discounting rates, and individual indifference points were inferred using the method described by [64], resulting in a single discounting rate (k-value) for each participant.

**Appetitive state.** To assess appetitive state at the time when the study was conducted, participants completed a 100pt visual analogue scale (VAS) rating of current hunger state ("Please rate how HUNGRY you feel right now") on a VAS end-anchored with "Not at all" and "Extremely". To reduce demand effects, this rating was embedded as one of seven mood and appetite ratings (the other items rated, but not used for analysis, were clear-headed, anxious, calm, tired, thirsty and full).

**Procedure.** Unlike Experiment 1, testing in Experiment 2 was conducted in person, with all testing in a quiet location, either in a test cubicle or office space, since the 4MT task cannot be conducted online. Data were collected between December 2017 and July 2018. After confirming consent, the measures were completed in the order: appetite and mood ratings, 4MT task, DDT, BIS11, EMQ, TFEQ. The mood scales and DDT, and the three questionnaire measures, were administered as surveys using Qualtrics software run on a PC laptop computer. Once all measures had been completed, participant age (years), height (m) and weight (kg) were recorded using a portable balance (Uten model EB5108H) and stadiometer. As in Experiment 1, participants were rewarded by having their name placed in a prize draw with a £25 prize.

**Data analysis.** As in Experiment 1, initial analyses used stepwise linear regression to model the various memory and impulsivity measures in relation to habitual diet. We first regressed each relevant memory and impulsivity measure against the DFS total score (Step 1), then added in, age, sex, and BMI (Step 2), TFEQ-R and TFEQ-D (Step 3) and rated hunger (Step 4). Note that for brevity we only report significant covariates: copies of the full models can be found in S3 Table.

To test the two models of how memory, impulsivity and diet scores were inter-related, we repeated the moderation analyses used in Experiment 1.

## Results

**Summary data and datachecks.** Three participants declined to provide height and weight data, and the results from two participants on the DDT task were uninterpretable, giving full data for 116 participants: analyses below include all available data for each measure. Summary data are shown in Table 4. Data from the EMQ, BIS11, 4MT and DFS distributed normally. As with Experiment 1, BMI and age were skewed by outliers, with three BMI scores in the obese range and five participants aged 38 or over were outliers: again we ran robust models when BMI and age were included. Outlier analysis also suggested three BIS11 scores were marginally outside the 95% confidence interval: as excluding data would impact power, we report analyses including these data but checked effects with robust models for BIS data.

**Memory measures.** Replicating Experiment 1, EMQ scores were related to DFS scores (Fig 4A: $r^2 = 0.16$, $F(1,118) = 21.92$, $p < 0.001$: DFS $b = 0.40$ [0.23, 0.56]). This effect was still significant when controlling for age, BMI, sex, TFEQD and TFEQR, none of which were significant predictors of EMQ. Similar effects were seen with the two sub-scales of the EMQ (Table 2, row 2).

The total correct responses (score out of 15) on the 4MT were also predicted by overall DFS scores: (Fig 4B: $r^2 = 0.04$, $F(1,118) = 4.43$, $p = 0.037$: DFS $b = -0.19$ [-0.37, -0.01]), with poorer memory performance associated with higher DFS scores. The coefficient relating 4MT and DFS remained significant first when age, sex and BMI were added, and when the TFEQ factors were added, although the model was no longer significant overall when all these factors were included.

**Impulsivity measures.** The model relating discounting (k) values from the DDT was not significant ($r^2 = 0.01$, $F(1,116) = 0.60$, $p = 0.44$), and DDT did not reliably predict DFS scores ($b = 0.07$ [-0.11, 0.26]). At step 3, DDT discounting values were positively associated with TFEQ ($b = 0.26$ [0.07, 0.45], $p = 0.007$), and negatively with TFEQR (0.27 [-0.46, -0.09], $p = 0.005$): age, bmi and sex were not related to DDT scores.

Replicating Experiment 1, the overall BIS11 score was predicted by the DFS total score (Fig 4C: $r^2 = 0.03$, $F(1,116) = 4.23$, $p = 0.042$, $b = 0.004$ [0.00, 0.01]), with no effect of age, sex, BMI or hunger. Adding in the TFEQ factors confirmed the effect of DFS ($b = 0.005$ [0.001, 0.009], $p = 0.015$) but also revealed a significant effect of TFEQD ($b = 0.022$ [0.006, 0.038]).

When relationships with BIS subscales and were examined (Table 3 row 2), after correction for multiple contrasts only the BISatt factor was significantly related to DFS scores.

**Mediation analyses.** As with Experiment 1, we also ran analyses to test the inter-relationships between impulsivity, memory and HFS intake. Because the combinations of memory and impulsivity measures resulted in eight different models, we have summarised these in Table 5 for concision. Since the DDT discounting (k) values did not correlate with the DFS scores, no mediation models could be fitted for that impulsivity measure. Initially we tested the vicious cycle model, predicting that the impact of diet on memory mediated the

**Table 4. Demographic characteristics, and hunger state at test, of participants in Experiment 2 (mean, SD and range).**

|  | n | Mean | Min | Max | SD |
|---|---|---|---|---|---|
| Age (years) | 120 | 21.95 | 18 | 41 | 3.32 |
| BMI (kg/m$^2$) | 117 | 23.19 | 17.2 | 37.9 | 3.52 |
| TFEQ-R | 120 | 7.7 | 0 | 21 | 5.1 |
| TFEQ-D | 120 | 7.0 | 1 | 15 | 3.5 |
| Rated hunger (0–100) | 120 | 41.1 | 0 | 100 | 28.2 |

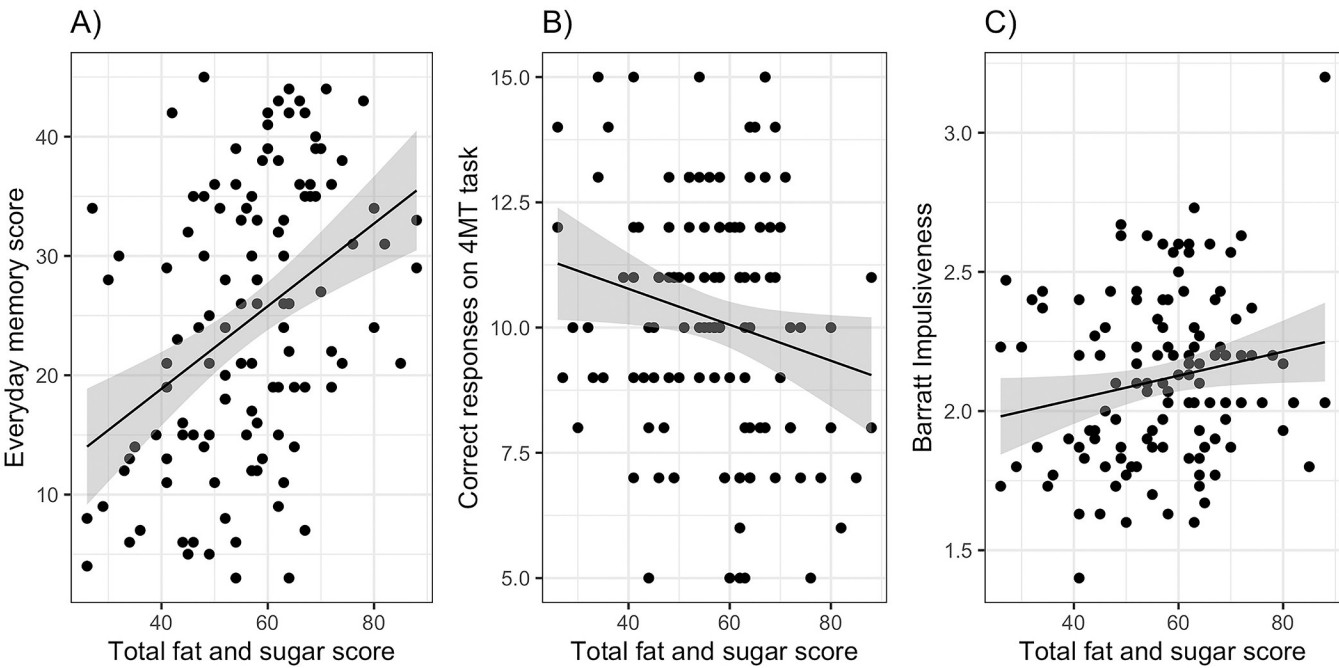

**Fig 4.** Memory and impulsivity measures related to overall fat and sugar intake (estimated using the DFS) from Experiment 2: (A) the Everyday Memory Questionnaire, (B) correct responses on the 4MT, (C) total score on the BIS11.

relationship between diet and impulsivity, and combining the different measures of memory and impulsivity. The initial model tested whether EMQ mediated the relationship between DFS and the BIS11 total score. However, the mediation model only found a significant relationship between EMQ and DFS scores (b = 0.346, 95% CI [0.213, 0.480], t = 5.07, p = <0.001): no pathway involving BIS11 total was significant. As the analysis of impulsivity identified BISatt as the only BIS11 factor related to DFS scores in this dataset, we therefore re-ran the mediation using BISatt. In that analysis the pathways between EMQ and BISatt (b = 0.008, 95% CI [0.001, 0.015], t = 2.30, p = 0.022), and EMQ and DFS (b = 0.346, 95% CI [0.213, 0.480], t = 5.07, p = <0.001) were significant, but the pathway between BISatt and DFS was not (b = 0.003, 95% CI [-0.005, 0.010], t = 0.63, p = 0.53), and the indirect effect linking DFS and BISatt through EMQ were not significant (b = 0.001, 95% CI [-0.002, 0.016], t = 0.62, p = 0.531).

For the 4MT task, the pathways between 4MT and BISatt (b = -0.024, 95% CI [-0.044, -0.004], t = 2.30, p = 0.022), and 4MT and DFS (b = -0.036, 95% CI [-0.068, -0.004], t = 2.19, p = 0.028) were significant, but the pathway between BISatt and DFS was not (b = 0.003, 95% CI [-0.001, 0.008], t = 1.43, p = 0.15), and the indirect effect linking 4MT and BIS through DFS (b = 0.001, 95% CI [0.000, 0.002], t = 1.82, p = 0.07) was not significant. As the VCM model would predict a significant indirect effect, and this was not significant in any of these models, these data did not support the VCM model.

We also ran the alternative mediation based on the trait model for both EMQ and 4MT memory tasks (lower rows in Table 5). With the EMQ there was some evidence to support the trait account, with higher scores on the BIS overall (b = 8.052, 95% CI [0.240, 15.87], t = 2.02, p = 0.043), and on the BIS attention subscale (b = 6.91, 95% CI [1.77, 12.05], t = 2.64, p = 0.008), predicting higher scores on the DFS, and then higher scores on the DFS predicting higher scores on the EMQ (i.e. more memory lapses: BIS total, b = 0.330, 95% CI [0.195, 0.464], t = 4.81, p = <0.001: BIS attention, b = 0.334, 95% CI [0.194, 0.475], t = 4.66, p =

**Table 5. Outcomes of the mediation analyses from Experiment 2 testing the vicious cycle and trait models of how diet, memory and impulsivity were inter-related.**

| Memory measure | Impulsivity measure | Vicious cycle model | | | | |
|---|---|---|---|---|---|---|
| | | Diet and memory (path a) | Memory and impulsivity (path b) | Indirect effect (path ab) | Diet and impulsivity (path c') | Total effect |
| | | Vicious Cycle Models | | | | |
| EMQ | BIS | 0.346 (0.213, 0.480) *** | (-0.002, 0.008) | 0.001 (-0.001, 0.003) | 0.003 (-0.002, 0.008) | 0.004 (0.000, 0.009) |
| EMQ | BISatt | 0.346 (0.213, 0.480) *** | 0.003 (-0.005, 0.010) | 0.001 (-0.002, 0.004) | 0.008 (0.001, 0.015) * | 0.009 (0.002, 0.016) |
| 4MT | BIS | -0.036 (-0.068, -0.004) * | -0.024 (-0.044, -0.003) * | 0.001 (0.000, 0.002) | 0.003 (-0.001, 0.008) | 0.004 (0.000, 0.009) |
| 4MT | BISatt | -0.036 (-0.068, -0.004) * | -0.032 (-0.065, -0.001) | 0.001 (0.000, 0.016) | 0.008 (0.001, 0.015) | 0.009 (0.002, 0.016) ** |
| | | Trait model | | | | |
| | | Impulsivity and diet (path a) | Diet and memory (path b) | Indirect effect (path ab) | Impulsivity and memory (path c') | Total effect |
| EMQ | BIS | 8.052 (0.240, 15.87) * | 0.330 (0.195, 0.464) *** | 2.655 (-0.287, 5.598) | 3.883 (-2.102, 9.87) | 6.538 (0.273,12.803) * |
| EMQ | BISatt | 6.910 (1.77, 12.05) ** | 0.334 (0.194, 0.475) *** | 2.310 (0.296, 4.325) * | 1.326 (-2.86, 5.51) | 3.636 (-0.888, 8.159) |
| 4MT | BIS | 8.052 (0.240, 15.87) * | -0.029 (-0.063, -0.005) | -0.235 (-0.574, 0.104) | -1.573 (-2.874, 0.272) * | -1.808 (-0.606, -1.808) ** |
| 4MT | BISatt | 6.910 (1.77, 12.05) ** | -0.028 (-0.061, 0.005) | -0.191 (-0.471, 0.088) | -0.899 (-1.84, 0.043) | -1.090 (-1.993, -0.187) * |

Data are coefficients and confidence intervals from each mediation model for each mediation pathway. S8 p<0.05

** p<0.01

*** p<0.001.

<0.001). However, while the indirect path was significant with BIS attention (b = 2.310, 95% CI [0.296, 4.325], t = 2.25, p = 0.025), this was not significant with the overall BIS score (b = 2.655, 95% CI [-0.287, 5.598], t = 177, p = 0.077).

With the 4MT task, higher BIS total scores predicted a lower 4MT score (b = -1.57, 95% CI [-2.874, 10.27], t = 2.37, p = 0.018), and BIS predicted DFS (b = 8.052, 95% CI [0.240, 15.87], t = 2.02, p = 0.043), but the path between 4MT and DFS was not significant (b = -0.029, 95% CI [-0.063, 0;005], t = 1.66, p = 0.097), nor was the overall indirect effect (b = -0.235, 95% CI [-0.574, 0.104], t = 1.36, p = 0.175), while with BIS attention, only the path between BIS attention and DFS was significant (b = 6.91, 95% CI [1.77, 12.95], t = 2.64, p = 0.008).

## Discussion

Experiment 2 replicated the associations between overall scores on the DFS and both the EMQ memory and BISatt impulsivity measures seen in Experiment 1, although the BISnp association did not replicate. Moreover, Experiment 2 also found evidence that poorer memory performance on a spatial memory task known to load heavily on hippocampal function, the 4MT task, was associated with higher scores on the DFS, albeit with a much lower effect size than was the EMQ. These associations were evident when age, sex, BMI, hunger state at time of testing, and TFEQ measures of eating attitude were controlled for.

Beyond HFS, Experiment 2 also provided further evidence that TFEQ scores correlate with impulsivity, but not memory, independently of habitual diet. Firstly, higher TFEQD scores were associated with both more impulsive responding on the DDT and higher scores on the BIS11 overall, and BISnp subscale in particular, replicating a previous study in this lab [26]. Secondly, we found that higher dietary restraint, measured as TFEQR scores, were associated with lower scores on the BIS1 overall, and BISatt subscale.

The mediation analyses testing the VCM did not replicate the suggestion from Experiment 1 that the relationship between impulsivity and diet was mediated by the effects of diet on memory. Critically, while diet and memory (EMQ and 4MT) were consistently related, impulsivity (BIS11 or DDT) was not related to memory (EMQ or 4MT), and the indirect pathway from diet to impulsivity via memory was not significant in any of the models. In contrast the trait mediation model did confirm pathways from impulsivity to diet and diet to memory for the BIS total and BIS attention measures, although outcomes with 4MT were less clear. Thus Experiment 2 tentatively suggests that the relationships between impulsivity and diet, and memory and diet, were independent, more in line with the trait than VCM model.

Overall, while the separate relationships between impulsivity and DFS scores, and DFS and memory scores, from Experiment 1 were at least partially replicated, the mediation analyses had a different outcome. Moreover, the effect sizes for the associations between DFS, memory and attention scores were small to moderate, and that could explain the differences in some outcomes from the two experiments. Experiment 3 therefore used a much larger sample, powered to provide a more robust test of these relationships.

The vicious cycle model (Fig 1B) suggests that impaired memory caused by habitual intake of a HFS diet leads to poor inhibitory control and that in turn exacerbates poor dietary choices [30,65]. As inhibitory control is a key facet of impulsivity [66], the rationale in Experiments 1 and 2 was that the impact of HFS on inhibitory control could explain the more general association of HFS and impulsivity [12], and both provided some evidence that both memory (EMQ and 4MT) and impulsivity (BIS11) were related to HFS intake. However, arguably a stronger test would be to use a measure of memory which included some aspect of inhibitory processing. We therefore also included the Remembering Causes Forgetting (RCF: [67]) task, which has not previously been used in studies examining effects of habitual diet. In the RCF task, rehearsed exemplars achieve enhanced retrieval while unrehearsed exemplars suffer impaired retrieval (e.g. [68,69]). Research shows that this effect involves activation of inhibitory control mechanisms which are recruited to overcome interference caused by competing memory traces [70]. We therefore hypothesised that habitual higher HFS diet would impair the ability to activate these inhibitory control processes, and so predicted less evidence of inhibition in the RCF task as a more direct test of the predictions from the vicious cycle model. Notably, one recent study contrasted performance on an RCF task in children and adolescents in relation to obesity. In line with our hypothesis, children with obesity showed no evidence of inhibition, but other groups did [71]. However, no dietary measures were taken in that study. Therefore Experiment 3 aimed to replicate and extend Experiments 1 and 2, adding in the RCF task as an additional measure of memory performance. Given the lack of evidence that the DDT measure of impulsivity was related to HFS intake in Experiment 2, and in the only published test of that relationship [12], impulsivity was assessed using the BIS11 only.

## Experiment 3

### Methods

**Design.**    The study examined the inter-relationships between impulsivity (measured by the BIS11), spatial memory (4MT), self-reported memory failure (EMQ) and performance on

**Table 6. Details of the stimuli used in the Remembering Causes Forgetting task from Experiment 3.**

| TREES | PROFESSIONS | METALS | WEAPONS | INSECTS | BIRDS |
|---|---|---|---|---|---|
| birch | author | aluminium | arrow | ant | blackbird |
| chestnut | baker | brass | bomb | beetle | dove |
| elder | dentist | copper | dagger | cricket | eagle |
| holly | farmer | iron | knife | fly | goose |
| maple | lawyer | mercury | pistol | locust | jackdaw |
| oak | nurse | nickel | rifle | mosquito | robin |
| yew | policeman | platinum | sword | scorpion | seagull |
| willow | sailor | tungsten | tank | wasp | wren |

the RCF task with self-reported consumption of a high fat and sugar diet (measured by the DFS questionnaire).

**Participants.** Two hundred and 86 participants (199 women and 87 men) participated, recruited as part of a larger study examining wider aspects of diet and taste preference. To meet the full study requirements, potential participants who had diabetes, had been diagnosed with an eating disorder, were taking prescription medication (other than oral contraception) or who were suffering from a respiratory illness were excluded. Since Experiment 2 only partially replicated Experiment 1, the effect size for the 4MT in Experiment 2 was small, and we had no prior data to power the study for the RCF task, we recruited a larger sample to ensure a more robust test. The study protocol was approved by the University of Sussex Science and Technology CRec ethics committee (approval ER/MARTIN/12).

**Materials.** Experiment 3 used the same versions of the DFS, BIS11, EMQ, 4MT, measure of appetitive state and TFEQ as Experiment 2. The RCF task was modelled closely on the procedure described in Anderson et al. [67], with the same four phases (learning, practice, distraction and surprise recall), but whereas the original was presented using a booklet, here all stimuli were presented on-screen (programmed using EPrime 1,2, Psychology Software). The stimuli (Table 6) consisted of six sets (categories) of eight words (exemplars). Four of the six categories (weapons, trees, metals and insects) were the same as in the original paper [67], but some of the exemplars were changed to ensure familiarity for a UK population. Because of the wider focus of the Experiment on diet, the food category was replaced with a new category of birds, along with an additional category of professions.

In the learning phase, all 48 items were presented in random order for 5sec each in the centre of the screen in the form "CATEGORY–EXEMPLAR", with an interval varying randomly between 1 and 2 seconds between stimuli. This was immediately followed by the rehearsal phase, which consisted of two blocks of 16 trials where four items from each of four of the categories (trees, professions, weapons and birds) were re-presented with the category name and first two letters of the relevant exemplar displayed. The task here was to complete the name of the exemplar, and participants had 8 seconds to do so, again with a delay of 1–2 seconds between stimuli: the 16 rehearsal items displayed in random order within each block. To distract participants from the stimuli in the RCF task, participants then completed the 4MT task (using the same procedure as Experiment 2): as they took different amounts of time to do, the task was followed by a further short variable delay so that the final recall phase of the RCF task started 10 minutes after the end of the rehearsal phase. For the surprise recall task, participants were given a sheet of paper with the instructions to write down as many of the stimuli as possible from the earlier word tasks.

**Procedure.** Testing took place in person in individual test cubicles at the Sussex Ingestive Behaviour Unit at the University of Sussex between 1000-1230h. To meet the requirements of

the broader study (which included a taste test), participants were instructed to consume their normal breakfast and then to abstain from eating, drinking (apart from water), smoking, chewing gum, or brushing their teeth for the two hours beforehand.

Participants completed a short mood and appetite questionnaire, including a rating of how hungry they were, at the start of the session, which was followed by the taste test (which was not relevant to these analyses). The memory tests started about 10 minutes after the hunger rating, with the learning and rehearsal phases of the computerised RCF task. On completion, they completed the 4MT task on an Apple iPad mini, followed by a short delay and the 3-minute surprise recall of the RCF task stimuli. Once finished, they completed the TFEQ, EMQ-R, DFS and BIS11 questionnaires in that order as a single online questionnaire administered using Qualtrics. Finally, their age and sex were recorded and height (SECA 220 stadiometer) and weight (SECA balance model 769) were measured to assess BMI.

**Data analysis.**   As in Experiments 1 and 2, principal data analysis used stepwise linear regression to model (Step 1) the various memory and impulsivity measures in relation to DFS scores while controlling for age, sex, BMI (Step 2), TFEQR and TFEQD (Step 3) and hunger at the time of testing (Step 4). For the RCF task, we included total number of items correctly recalled as a simple measure of short-term memory and then the ratio of total unrehearsed to rehearsed items recalled as a measure of memory inhibition (where a higher ratio would indicate weaker inhibition: [67]). As neither Experiment 1 or 2 found differences between EMQ subscales, we saw no purpose in repeating those analyses, but did re-examine relationships with the BIS subscales.

We again used mediation to test the alternative pathways relating diet, memory and inhibition using the same R packages as before.

## Results

**Summary data and datachecks.**   An online error meant no DFS data were recorded for one male participant, and consequently his data were excluded. Issues with the iPad meant 4MT data were missing for four participants. Demographic data (age, height and weight) were missing for a further ten participants: summary data are shown in Table 7. All available data were used in analyses to maximise power. Data were also checked for normality and outliers: 4MT and BMI were both significantly skewed and we therefore used robust models for those analyses.

**Memory measures.**   Overall, there was evidence that higher habitual intake of a high fat sugar diet was associated with poorer response on all three memory tasks. Regression again found EMQ scores to be related to the overall DFS score (Fig 5A: $r^2 = 0.09$, $F(1,283) = 26.24$, $p<0.001$: DFS $b = 0.29$ [0.18, 0.40]), and this remained true when controlling for age, sex, bmi and hunger: sex was a significant predictor ($b = 2.75$ [0.57, 4.93], $p = 0.014$), with male participants having higher EMQ scores. At step 3, TFEQD was also related to the total DFS score

**Table 7. Demographic characteristics, and hunger state at test, of participants in Experiment 3 (mean, SD and range).**

|  | n | Mean | Min | Max | SD |
|---|---|---|---|---|---|
| Age (years) | 275 | 21.4 | 18 | 34 | 2.7 |
| BMI (kg/m$^2$) | 275 | 22.4 | 16.0 | 39.4 | 4.7 |
| TFEQ-R | 285 | 7.5 | 0 | 19 | 4.7 |
| TFEQ-D | 285 | 7.0 | 1 | 15 | 3.2 |
| Rated hunger (0–100) | 285 | 47.5 | 0 | 100 | 25.7 |

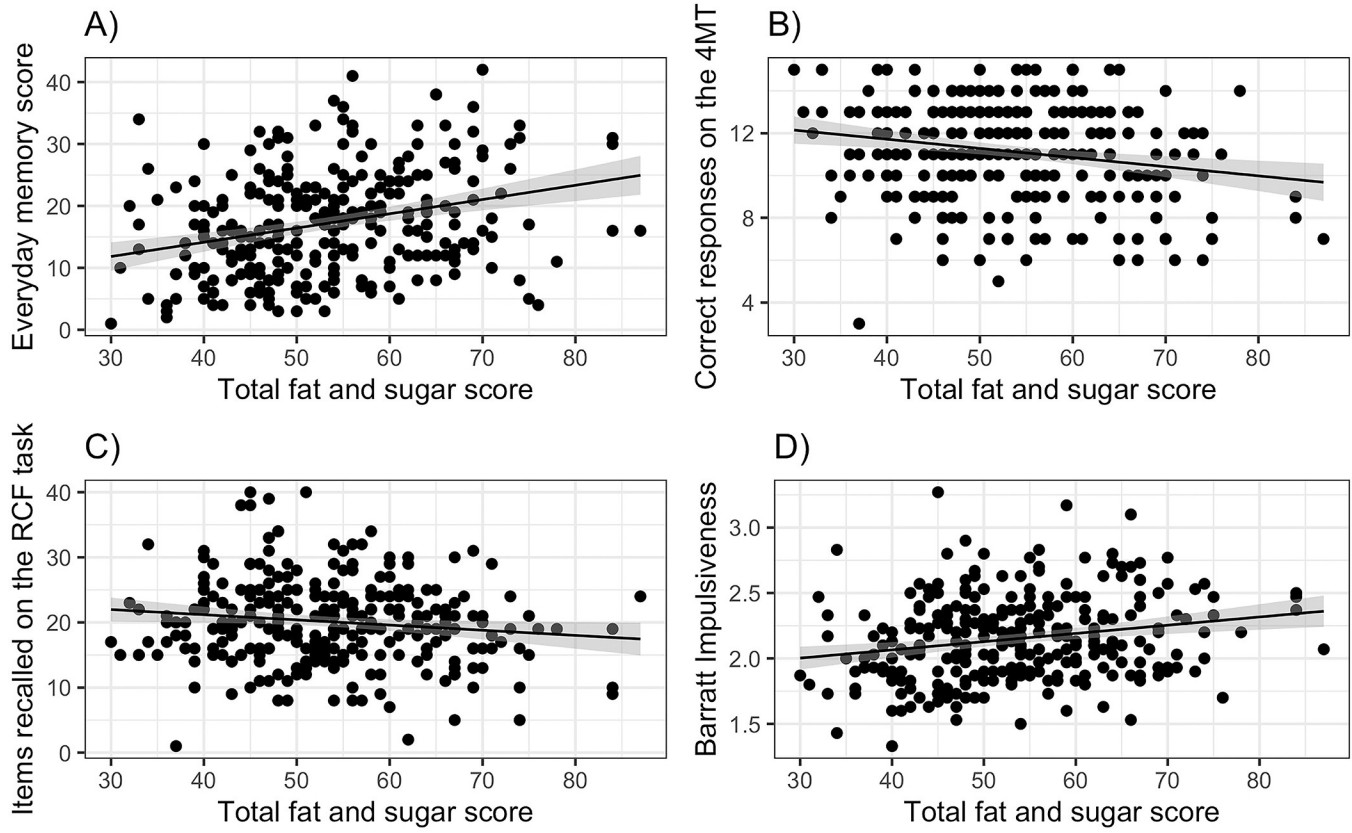

**Fig 5.** Memory and impulsivity measures related to overall fat and sugar intake (estimated using the DFS) from Experiment 3: (A) the Everyday Memory Questionnaire, (B) correct responses on the 4MT, (C) total items recalled on the RCF task, (D) total score on the BIS11.

($b$ = 0.65 [0.31–0.99], p < 0.001), but EMQ remained a significant predictor ($b$ = 0.17 [0.08, 0.27], p < 0.001).

Higher DFS scores were also associated with poorer performance on the 4MT task (Fig 5B: $r^2$ = 0.04, F(1,279) = 11.36, p<0.001: DFS $b$ = -0.05 [-0.08, -0.02]), and this remained significant when age, sex, bmi, hunger, TFEQD or TFEQH were added: these analyses also found that performance on the 4MT task depended on sex ($b$ = 0.65 [-0.28, 1.58], p = 0.001), with male participants performing better than female. For the RCF task, overall recall decreased with increasing DFS score (Fig 5C: $r^2$ = 0.02, F(1,283) = 4.95, p = 0.027: DFS $b$ = -0.13 [-0.25, -0.02]): adding in age, sex, BMI and hunger did not improve the fit but TFEQD was also associated with weaker overall recall ($b$ = -0.30 [-0.58, -0.03]). However, the measure of relative memory inhibition was unrelated to DFS ($r^2$ < 0.01, F(1,283) = 0.24, p = 0.63: DFS $b$ = -0.03 [-0.15, 0.09]).

**Impulsivity measures.** As before, the overall BIS11 score was positively associated with the DFS total score (Fig 5D: $r^2$ = 0.05, F(1,283) = 13.29, p <0.001, b = 0.21 [0.10, 0.33]). Neither age BMI or hunger were significant, but TFEQD was positively associated with DFS ($b$ = 0.027 [0.014, 0.039]). In this larger sample, both BISatt and BISmot were significantly relate to DFS score, albeit with small effect sizes (Table 3, row 3).

**Mediation analysis.** We first tested the VCM. In line with Experiment 1, and in contrast to Experiment 2, BIS11 and EMQ scores were related in Experiment 3 (r(285) = 0.54, p = < .001, 95% CI = [0.45, 0.62]). Without accounting for EMQ, there was a significant direct relationship between DFS and BIS11 (b = 0.006, 95% CI [0.003, 0.01], t = 3.72, p = <0.0001).

When EMQ was included, that direct effect was no longer significant (b = 0.002, 95% CI [-0.001, 0.005], t = 1.21, p = 0.226), but the indirect effect was significant (b = 0.004, 95% CI [0.002, 0.007], t = 4.20, p < 0.001: Fig 6A). Therefore, the relationship between DFS and BIS11 scores appeared to be mediated by EMQ. In contrast, the relationship between correct responses on the 4MT task and BIS11 was not significant (r(281) = -0.03, p = 0.60, 95% CI = [-0.31, 0.09]), and therefore memory measured by the 4MT could not explain the significant relationship between DFS and BIS11 scores.

We next tested how total recall from the RCF task was related to impulsivity and DFS scores. Initial analyses found significant correlations between the total recall score from the RCF and both the total DFS score (r(285) = -0.13 [-0.24, -0.01], p = 0.033) and overall score on the BIS11 (r(285) = -0.14 [-0.25, -0.03], p = 0.033), and so the conditions to conduct mediation analysis were met. Mediation confirmed a direct relationship between DFS and BIS11 when RCF was accounted for (b = 0.006, 95% CI [0.003, 0.010], t = 3.72, p < 0.001), but the indirect effect was not significant (b = 0.000, 95% CI [0.000, 0.001], t = 1.51, p = 0.13) giving no evidence that memory as measured by RCF mediated the relationship between BIS11 and DFS. The measure of relative memory inhibition from the RCF task did not correlate significantly with either diet (r = -0.02, p = 0.716) or BIS11 total (r = -0.11, p = 0.105), therefore a mediation model with the inhibition model could not be justified.

Finally, we tested the trait model for each memory measure. For EMQ, there was a significant direct relationship between DFS and BIS11 both when EMQ was (b = 13.37, 95% CI [10.90, 15.84], t = 10.62, p = <0.0001) and was not included (b = 14.41, 95% CI [11.86, 16.96], t = 11.09, p = <0.0001), but also a weak indirect effect (b = 0.41, 95% CI [0.231, 1.853], t = 2.52, p = 0.012: Fig 6B). In contrast, for the 4MT, there was no significant direct relationship between DFS and BIS11 when EMQ was not included (b = -0.229,95% CI (-1.05–0.59), p = 0.547), but DFS and BIS11 were related when EMQ was included (b = 7.052, 95% CI [3.201, 10.904], t = 3.59, p = <0.0001), and there was a significant indirect effect (b = -0.309, 95% CI [-1.051–0.593], t = 2.38, p = 0.018). Finally, for the RCF total measure, there was a significant direct relationship between DFS and BIS11 when EMQ was (b = 7.136, 95% CI [3.309, 10.963], t = 3.65, p = <0.0001) and was not included (b = -2.93, 95% CI [-5.247, 16.96], t = 11.09, p = <0.0001), but no indirect effect (b = -0.456, 95% CI [-1.010, 0.099], t = 1.62, p = 0.108).

## Discussion

Experiment 3 provided further evidence that higher intake of fat and sugar, indexed using the DFS, was related with poorer memory and higher impulsivity. With memory, the relationship between DFS and EMQ seen in the two previous experiments was replicated, as was the weaker but significant relationship between DFS and performance on the 4MT task. DFS was also weakly related to the overall number of items recalled in the surprise recall phase of the RCF task, but there was no evidence that higher DFS scores were associated with weaker inhibitory effects of rehearsal on recall. With impulsivity, the overall BIS11 score was related to DFS, again replicating the previous studies, and this again was due to scores on the BIS attention subscale, with no evidence of relationships with the motor or non-planning subscales.

Finally, when we used mediation to explore the inter-relationships between diet, memory and impulsivity, the outcome depended on which measure of memory was tested in the model. With EMQ, there was some evidence to support the VCM model that the relationship between DFS and BIS11 was at least in part mediated by the effects of DFS on memory, but no such mediation was seen with 4MT performance, which was unrelated to BIS11, or for the RCF measure, where the significant relationship between BIS11 and DFS was not mediated by

## A) Vicious Cycle Model

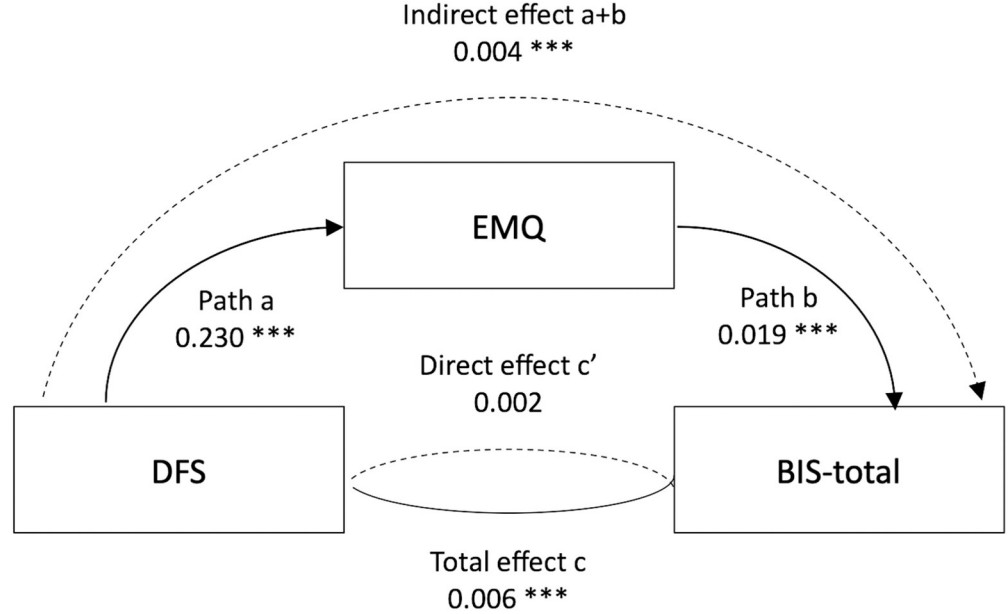

## B) Trait Model

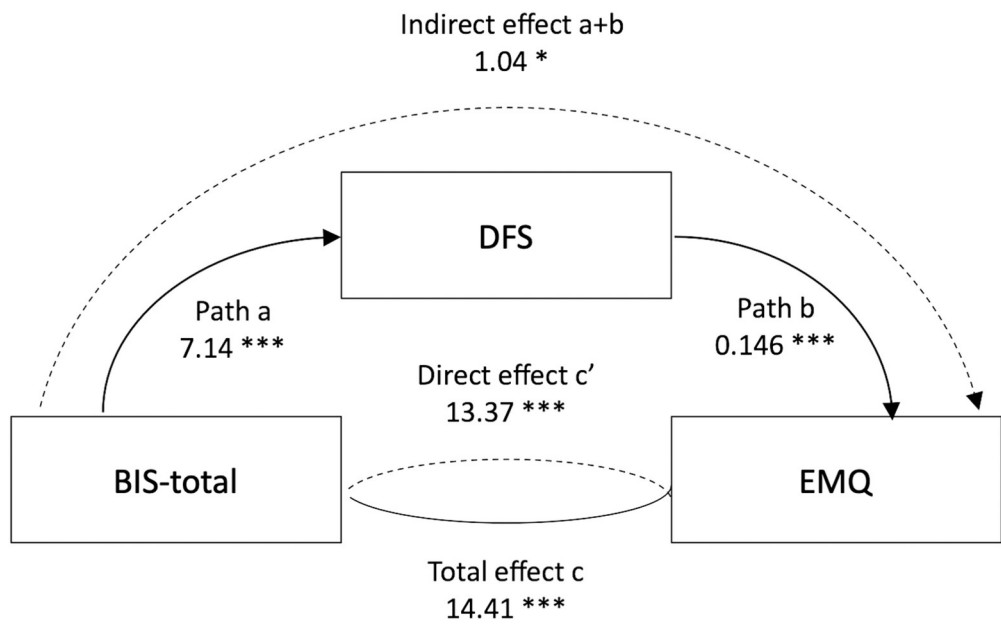

**Fig 6.** Summary diagrams of (A) the vicious cycle and (B) trat mediation models relating fat and sugar intake measured using the DSF, memory (Everyday Memory Questionnaire, EMQ or Four Mountains task, 4MT) and the total score on the Barratt Impulsiveness Inventory (BIS11). The pathways are labelled with the standard mediation coefficient labels, with the b-values for each relationship. * p< 0.05, ** p<0.01, *** p<0.001.

RCF scores. For the trait model, there was also evidence that high scores on the BIS predicted DFS scores, and a significant indirect effect suggested that in turn influenced EMQ, but no such indirect effects were seen with the 4MT or RCF memory measures.

Importantly, none of these modelled relationships were indirectly caused by differences in BMI, age, sex, hunger at the time of testing, or by TFEQ scores. With TFEQ we did find some significant relationships: the finding that TFEQD was related to BIS11 replicated the earlier experiments and wider literature [26,27], but the finding that TFEQD was also associated with poorer memory on both the EMQ and RCF measures was unexpected. There were also some effects of sex in these data, with male participants reporting more memory problems based on responses on the EMQ, but better performance on the 4MT.

## General discussion

Across the three experiments, we found a consistent relationship between self-reported everyday memory failures, as indexed by the short-form EMQ measure, and habitual intake of fat and sugar as measured by the DFS. Additionally, in Experiments 2 and 3 the association between fat and sugar intake and an objective measure of memory, performance on the hippocampally sensitive 4MT task, was also seen albeit with a smaller effect size, while in Experiment 3 overall recall of words from the RCF task was also negatively related to DFS scores. Thus all three Experiments provided further evidence that habitual intake of fat and sugar was associated with impaired memory, adding significantly to the existing published evidence both from similar correlational studies [10,11,17,72] and studies using short-term overexposure to fat and sugar [18,19]. Those outcomes are further supported by longitudinal studies that also tend to show negative effects of habitual fat and sugar on memory, albeit with more variability (reviewed in 5). Overall, although the size of effects was generally small, we found remarkable consistency in the negative relationship between habitual diet and both self-report and objective memory performance.

The current paper adds to the small literature on how habitual intake of fat and sugar relates to impulsivity. Here there were also clear and replicable findings. Overall impulsiveness measured by the BIS was associated with higher scores on the DFS in all three experiments, with the BIS attention and motor sub-scales the most consistent, replicating the first experiment in the paper by Lumley et al. [12]. The finding that the Dickman dysfunctional, but not functional, impulsivity measure was also related to higher scores on the DFS was unsurprising, given that the dysfunctional measure is usually correlated with higher BIS scores [52,73], as was the case in Experiment 1 here. This implies that the nature of impulsivity relating to intake of a diet higher in fat and sugar is rash decision making, in keeping with the wider associations of rash decision making with binge-eating [74,75], alcohol and substance misuse [76,77]. With binge-eating, there is strong evidence that this rash decision making reflects a lack of inhibitory control combined with increased food-specific reward sensitivity (reviewed in [78]). However, not all studies that have examined impulsivity in relation to habitual intake of fat and/or sugar have found effects. Notably, a recent also used the DFS measure to explore the relationship between habitual diet, memory and impulsivity [79]. In that study, performance participants scoring less than 52 or more than 62 on the DFS completed a series of tasks and measures including the UPPS and Behavioral Inhibition and Behavioral Activation System

Scales (BIS/BAS) [80], and found no group differences in impulsivity. That study built on an earlier investigation using a similar between-groups approach [81], again including the UPPS and BISBAS measures, and also found no differences between high and low DFS groups. We do note that the sample sizes (19 in [81], 60 in [79]) in those two studies were lower than in the current experiments, but the lack of significance does warrant caution in interpreting the strength and reliability of how habitual diet and impulsivity are inter-related. Taking a slightly different approach, Steele et al. [82] assessed recent dietary intake using three 24-hour recalls in the week prior to testing, and relating that to both a food-related and monetary delay discounting measures. They found no relationship between dietary fat and impulsive choice, in line with the lack of association between DFS and delay-discounting in Experiment 2 here, but interestingly did find that greater intake of dietary sugar was associated with greater delay and magnitude sensitivity in the food-related impulsive choice task (that is they preferred to wait for a larger reward than accept the smaller immediate reward). That finding fits with the association between DFS and impulsive choice measured using a food-related delay-discounting task [12]. It also suggests that it would have been useful to have included a similar measure in the current studies. We had also considered analysing our current studies separately for estimated dietary fat and sugar intake, but the DFS is not designed to achieve this (although it is possible to generate separate measures for fat, fat with sugar and sugar from the DFS, the number of items for sugar is very small, and so any lack of significance could arise through restricted variance).

In the studies reported here, the most consistent relationship between impulsivity and DFS scores were seen with the BIS attention sub-scale (see Table 3), and some speculation about why that was the case is warranted. It is first notable that other studies that have examined how the BIS sub-factors relate to other measures of eating behaviour have also noted stronger associations with the BIS attention than motor or non-planning sub-scales. In particular, higher BIS attention scores have been reported to relate to different measures related to overeating: the disinhibition scale from the TFEQ [83], measures of how pleasurable it would be to consume a variety of high-fat foods [84]. But not all studies see these effect: all three BIS sub-scales were similarly related to TFEQ disinhibition and actual short-term intake in a recent study [85], and studies in our lab have found that the motor rather than attentional sub-scale was more strongly related to disinhibition [26,86]. There have also been reports of higher scores on the BIS attention subscale for people with binge-eating disorder (e.g. [87–89]). This suggests that the BIS attention, also referred to as cognitive flexibility, is associated with greater risk of overeating and unhealthy diet choice. Foods which are higher in fat and/or sugar haven long been interpreted as being more rewarding (e.g. [90,91]). BIS attention has also been found to be associated with measures of food reward and addiction (see [92]). Thus of the impulsivity measures we included, BIS attention may have been the measure best able to detect trait reward sensitivity in this population, and this suggests that future studies should explore other measures which assess general reward sensitivity.

Inhibitory failure is also a key feature of the vicious cycle model of obesity [30]. That model suggests that it is a failure to inhibit retrieval of food-related memories as a function of acute appetitive state that is a critical component of the perpetuation of unhealthy diet choices. If that explanation for the relationship between HFS intake and impulsivity is correct, then we predicted that memory mediates the HFS–impulsivity relationship. There was some evidence in the present studies to support that contention. In both experiments 1 and 3, mediation analysis found evidence of a significant indirect relationship between DFS and the total BIS score mediated through the relationships between DFS and EMQ, and EMQ and BIS. The clearest effect was seen in Experiment 3, which used a larger sample to increase the power to detect this effect. This suggests that impulsivity is increased by having a habitual higher intake of an

unhealthy diet. However, since we did not find the same in Experiment 2, possibly through lack of study power, this conclusion is made with some caution and suggests that further investigation is needed. In this context, in Experiment 3 we used the RCF task to try and measure a memory-related hippocampally-sensitive measure of inhibition. However, the inhibition measure from the RCF was not associated with higher DFS scores but total recall on that measure was. Thus the RCF data further evidenced a wider association between higher DFS scores and poorer memory, but did not find specific evidence for greater inhibition. In the present studies we explored whether that idea could be expanded to incorporate wider inhibitory control processes associated with the hippocampus, and the lack of evidence for this in the RCF task suggests that is not the case. That finding doesn't in itself challenge the VCM model, but does question whether the general effects on memory we consistently found can be attributed to inhibitory failure.

We also explored an alternative conceptualisation of the relationship between impulsivity and higher HFS intake where trait impulsivity increases the risk of making unhealthy diet choices, and those choices then impact memory without altering impulsivity (the trait model: Fig 1A). Here the suggestion is that impulsivity precedes the cognitive impairment induced by unhealthy diets. There was again some evidence in the present data to support that model: in Experiment 2 there was an association between impulsivity (measured using the BIS) and higher scores on the DFS, but no association with EMQ, and the mediation of the relationship between DFS and BIS in Experiment 1 was only partial. Likewise, in Experiment 3 there was evidence for an indirect pathway between the BIS total score and EMQ (but not 4MT or RCF) memory measure.

Overall, the mediation analyses do not provide clear strong support for the VCM or trait accounts of the inter-relationships between diet, memory and impulsivity. A better conceptualisation of the inter-relationships between impulsivity, unhealthy diet and memory might be that certain aspects of impulsivity, especially the risky decision making measured by BIS, increases the risk of unhealthy diet choice which then impacts reward sensitivity through the role of the hippocampus as modulator of food-based memory [36]. The main brain area classically associated with impulsivity is the prefrontal cortex [93,94], whereas the hypothesised impacts of a HFS diet are through the hippocampus [19]. A wealth of data (reviewed in [65]) suggest that the hippocampus has a particular role in integrating interoceptive cues relating to appetitive state which act to modulate the affective quality of food-related memories, so reducing the perceived reward value of the remembered food. Thus the negative impact of HFS on the hippocampus may further enhance the attraction of unhealthy food by reducing the modulating impacts of interoception.

A further consideration is whether the effects of HFS diets are limited to the hippocampus. In particular, some studies in rodents have suggested that HFS diets also impact the prefrontal cortex [95–97]. As the prefrontal cortex is heavily implicated in impulsivity, an alternative interpretation of the association between DFS scores and measures of impulsivity in the present studies could be that unhealthy diet directly effects the prefrontal cortex as well as the hippocampus. However, a recent investigation of striatal and prefrontal function using a working memory task found no association of DFS with behavioural or neural measures of PFC function [79].

In the present studies we found no evidence of any effects of body-size on memory or impulsivity measures, in contrast to studies clearly showing impaired memory relating to obesity (e.g. [98,99]). However, all three studies here used a relatively young population with low levels of obesity: indeed, in the study by Cheke et al. [41] that reported reduced memory in patients with obesity, their obese group had BMIs in the high 30's, whereas only 5 of the 495 participants in the three studies here had BMI greater than 35. It would therefore be important

to examine further the relationships between diet, memory and impulsivity in a sample with much higher representation of obesity. However, the VCM suggests that obesity is a consequence of the sustained negative impact of HFS consumption, and finding these relationships in a larger normal weight younger population suggests that those in the study population consuming a more unhealthy diet are at risk of future obesity.

One novel finding in the current studies was evidence of impaired spatial memory, measured using the 4MT, associated with higher DFS scores. Spatial learning and memory have been widely documented to be impaired in rodents consuming HFS diets (reviewed in 3), but the present findings appear to be the first to see this using a hippocampal-sensitive spatial memory task in humans. It would be interesting to explore these effects further, in particular focussing on spatial tasks that include some aspect of food-related foraging.

One weakness in the present studies was our reliance on a self-report food frequency measure to estimate long-term habitual intake of fat and sugar. While the measure we used has been validated for Western diets [46], it should be regarded as a measure of relative rather than absolute intake. The consistency within the current findings does suggest that these measures are reliably measuring an important aspect of diet, but further replication using more precise nutritional tools would be valuable for future studies. However, the DFS does focus on saturated fat and free sugar: the items do not include sugar from fruit, and only includes oil under cooking fat. The three experiments reported here also all focus on general measures of memory and impulsivity. Although we had a theoretical rationale for doing so, a further limitation is that we could not explore effects which were specific to food, either as a reward or in relation to impaired food-related memory. In relation to impulsivity, the omission of a food-related delay-discounting task limited our interpretation of how diet and impulsivity were inter-related and warrants further investigation. We also note that E1 used an entirely female population, and while we did not restrict recruitment to females only for E2 and E3, men were under-represented in both studies, reflecting the wider population we recruited from, but this limitation suggests future studies should examine more specifically the possibility of sex-based influences on these relationships.

Overall the present studies set out to extend recent work on the negative impacts of habitual intake of unhealthy diets on memory and impulsivity, and these cross-sectional studies provided consistent evidence that higher HFS intake was associated with poorer memory and increased impulsivity. These studies also provided some support for the vicious cycle model [30], but also suggest trait impulsivity may be a risk factor for unhealthy diet choice which then becomes entrenched by the negative effects of the selected diet.

## Supporting information

**S1 Table. Correlations between the different impulsivity measures in Experiment 1.**
(DOCX)

**S2 Table. Outcomes of the mediation analyses from Experiment 1 testing the vicious cycle and trait models of how diet, memory and impulsivity were inter-related.** Data are coefficients and confidence intervals from each mediation model for each mediation pathway. S8 $p<0.05$, ** $p<0.01$, *** $p<0.001$.
(DOCX)

**S3 Table. The main regression models from Experiment 2.** EMQ, Everyday Memory Questionnaire: 4MT, total score on the Four Mountains Task: BIS total, total score on the Barratt Impulsiveness Scale: ddt k, k-value from the delay discounting task: DFS, scores on the Dietary Fat and Sugar questionnaire: BMI, body mass index: TFEQR, restraint scale from the Three

Factor Eating Questionnaire: TFEQD, disinhibition scale from the Three Factor Eating Questionnaire.
(DOCX)

## Acknowledgments

The authors wish to thank the following undergraduate and masters students at University of Sussex who assisted with data collection: Experiment 1, Laura Boyd; Experiment 2, Molly Salisbury and Ken Kilsby; Experiment 3, Selen Atak, Freya Pugh and Jana Rohregger. MY designed the studies, oversaw data collection for Experiment 1 and 2, conducted the analysis and was the primary author. RAr oversaw data collection for Experiment 3 and contributed to manuscript writing. RAt provided advice and support for conduct of the meta-analyses. RS and HF provided intellectual input into the study design and conduct, and commented on early drafts of the MS.

## Author Contributions

**Conceptualization:** Martin R. Yeomans, Heather Francis, Richard J. Stevenson.

**Data curation:** Martin R. Yeomans, Rhiannon Armitage, Rebecca Atkinson.

**Formal analysis:** Martin R. Yeomans, Rebecca Atkinson.

**Methodology:** Martin R. Yeomans, Rhiannon Armitage, Richard J. Stevenson.

**Project administration:** Martin R. Yeomans, Rhiannon Armitage.

**Software:** Rebecca Atkinson.

**Supervision:** Martin R. Yeomans, Rhiannon Armitage.

**Visualization:** Martin R. Yeomans.

**Writing – original draft:** Martin R. Yeomans.

**Writing – review & editing:** Rhiannon Armitage, Rebecca Atkinson, Heather Francis, Richard J. Stevenson.

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
