## [Decision Letter · Decision Letter 0]

29 May 2023

PONE-D-23-04749Habitual intake of fat and sugar is associated with poorer memory and greater impulsivity.PLOS ONE

Dear Dr. Yeomans,

Thank you for submitting your manuscript to PLOS ONE. After careful consideration, we feel that it has merit but does not fully meet PLOS ONE’s publication criteria as it currently stands. Therefore, we invite you to submit a revised version of the manuscript that addresses the points raised during the review process. We received two reviews from expert reviewers who produced excellent work with detailed points for improvement of your manuscript. Please, recognized the hard work they have done for making your manuscript more impactful for the readers. See guidance for the revision process below and we hope that you find the comments of the reviewers supportive.  Please submit your revised manuscript by Jul 13 2023 11:59PM. If you will need more time than this to complete your revisions, please reply to this message or contact the journal office at plosone@plos.org. Please include the following items when submitting your revised manuscript:A rebuttal letter that responds to each point raised by the academic editor and reviewer(s). You should upload this letter as a separate file labeled 'Response to Reviewers'.A marked-up copy of your manuscript that highlights changes made to the original version. You should upload this as a separate file labeled 'Revised Manuscript with Track Changes'.An unmarked version of your revised paper without tracked changes. You should upload this as a separate file labeled 'Manuscript'.

We look forward to receiving your revised manuscript.

Kind regards,

Hans-Peter Kubis, PD. Dr. rer. nat.

Academic Editor

PLOS ONE

3. Please include a copy of Tables 8  and 9 which you refer to in your text on pages 29 and 32.

Reviewers' comments:

Reviewer's Responses to Questions

**Comments to the Author**

1. Is the manuscript technically sound, and do the data support the conclusions?

Reviewer #1: Partly

Reviewer #2: Yes

2. Has the statistical analysis been performed appropriately and rigorously? 

Reviewer #1: Yes

Reviewer #2: Yes

3. Have the authors made all data underlying the findings in their manuscript fully available?

Reviewer #1: Yes

Reviewer #2: Yes

4. Is the manuscript presented in an intelligible fashion and written in standard English?

Reviewer #1: Yes

Reviewer #2: Yes

5. Review Comments to the Author

Reviewer #1: The manuscript is technically sound, the experiments were conducted rigorously, and the sample sizes are sufficient and justified by power analyses. Most of the conclusions drawn from the data are appropriate and do not overreach. Nevertheless, some specifications would improve the understanding of how the data was acquired and conclusions drawn.

• line 238: How much time passed between the assessment of the TFEQ data and DFS, EMQ, BIS11, and DI data? How could this delay affect the association between the two data sets?

• line 245: please provide or reference the standard formula for BMI

• line 250: the authors mistyped “regression”

• line 319: What are the “different dietary measures” the BIS11 sub-scales were related to?

• line 320: I invite the authors to precisely report the statistics for the association of BIS sub-scales with DFS and refrain from generalized statements (“to some extend related”), especially when the association of BISmot with DFS did not reach significance.

• line 329: The authors mistyped the alpha level (p < 0.10 instead of p < 0.01)

• line 400: can the authors explain why they deemed controlling for hunger at the time of testing necessary? Is there any literature linking hunger state to their measures of impulsivity or memory? Were all memory and impulsivity measures controlled for hunger? The manuscript only mentions BIS11 (line 483) but not DDT (line 480) or EMQ (line 462)

• I invite the authors to address in the discussion how they interpret the finding that BISatt was most consistently associated with HFS and involved in the mediation analysis to test the two possible models? Can they speculate why this sub-component of impulsivity, in contrast to the others, is involved in HFS intake and hippocampus-related memory?

• line 603: When did the taste test in experiment 3 happen and how much later were hunger rating assessed?

• line 616: Could the authors explain why a higher ratio of total unrehearsed to rehearsed items recalled can be regarded as a measure of memory inhibition?

• line 696: the authors mention evidence that habitual intake of fat, alone or in combination with sugar, underlies the association of DFS scores with memory and impulsivity. Which analyses provide this evidence? Please mention in the statistical analysis section that models with the DFS sub-scores were conducted and report the statistical findings in the results section. Were the sub-scores also used for the mediation analyses? If not, the authors should consider these analyses. And were these analyses done only for experiment 3? Furthermore, the text mentions tables 8 and 9, which are not included in the manuscript.

• line 730: the authors state that to their knowledge the present manuscript is the second to examine how habitual intake of fat and sugar relate to impulsivity. In fact, at least three further studies investigated this relationship: 1) Steele et al., Appetite, 2021 (DOI: 10.1016/J.APPET.2021.105292) report no association of fat but sugar with impulsive choices, 2) Hartmann et al., J Neuroendocrinology, 2020 (DOI: 10.1111/JNE.12917), found no association of two extreme groups of low and high consumers of fat and sugar diet with the UPPS, 3) Hartmann et al., Appetite, 2023 (DOI: 10.1016/J.APPET.2023.106477) found no association of two extreme groups of low and high consumers of fat and sugar diet with the BIS15

• line 737: the authors mention the correlation of BIS11 with the Dickman dysfunctional scale. Please mention this test in the statistics and results section

• line 742: the authors state that inhibitory failure is a key feature of the vicious cycle model. Could the authors discuss why the RCF task ratio, which is supposed to measure inhibitory control, is not associated with HFS in contrast to other measures of memory?

• line 781: Hartmann et al., Appetite, 2023 (DOI: 10.1016/J.APPET.2023.106477) investigated the association of habitual fat and sugar intake with striatal and prefrontal function using a working memory task. They did not find any association of HFS with behavioral or neural measures of PFC function.

• Figure 1a shows a feedback loop between HFS and memory, but this feedback is never mentioned in the text introducing the trait model. I strongly recommend removing this feedback loop and only show a linear relation with effects of HFS on memory in line with the background provided in the text.

• In various occasions throughout the manuscript the authors mention moderation analysis although I assume they meant mediation.

• The causal relation between impulsivity, HFS, and memory for the trait model is not always presented in a coherent way. In line 131 the authors write that trait impulsivity is associated with higher HFS intake, which in turn leads to impaired memory. But in line 146 the authors state that the trait model does not make any suggestions about any negative consequences of HFS intake. In the same sentence the authors claim again that HFS might influence memory, but it would be independent from impulsivity. I invite the authors to explicitly state the relationship of factors in the trait model and converge this on to the mediation analyses, describing how significance in which pathway is in favor of the trait model.

• In my opinion, the authors should be more careful with the conclusion they can draw from their data about the vicious cycle model and mention some limitations of the experimental design. The vicious cycle model proposed by Hargrave, Jones & Davidson (2017) and Davidson et al. (2019) mentions food cue reactivity as the main link between impaired memory inhibition, and though impulsivity and food cue reactivity can be related, van den Akker et al. (2014) recommends a multimodal assessment of both constructs. Furthermore, the vicious cycle model relies on food-related memories and memories of post-ingestive rewards. The measures of memory presented in this manuscript pertain to everyday memory (EMQ), spatial memory (4MT), and short-term memory (RCF task). I invite the authors to consider these limitations and discuss if and how the presented measures relate to food cue reactivity and food memory and what conclusions can be drawn in favor of the vicious cycle model.

Overall, the authors chose the appropriate statistical models to answer their research questions. The statistical analysis has been conducted rigorously, though it is not always clear to the reader how exactly the models were specified and what is the entirety of models that were run by the authors. I advise the authors to provide a section explaining all the models that were run in such a form that it becomes clear which variables were used as independent variable, dependent variable, and covariate (dv ~ iv + covar). This can be shown in a general manner and then specified for the respective variables for each experiment. Furthermore, it would be beneficial to showcase the mediation models in the conventional graph, depicting the predictor, mediator, and outcome variable as well as direct and indirect effects. The same figures should be used to explain the results of the mediation analyses, showing a and b for the two paths of the indirect effect, the indirect effect ab, the direct effect c’, and the total effect c.

• How did the authors define statistical outliers in their data? Were all participants with BMI > 30 statistical outliers or excluded based on their body weight status obese?

• line 264: The authors seem to have misspecified the mediation model to test the vicious cycle model (“any effect of memory on DFS would be mediated by impulsivity”). In line 334 they specify the model to test the vicious cycle model as “whether EMQ mediated the relationship between DFS and BIS11”

• line 279: For other researchers being able to replicate the authors’ analyses I recommend describing how the robust models were different to the conventional models.

• line 253: Following up on my general notion that the manuscript would benefit from a clear description of the statistical models, I invite the authors to explain whether in step 3 TFEQ was added to the model from step 1 or step 2.

• line 319: participants with obesity were excluded from the analysis of the BIS11 sub-scales but a similar procedure is not mentioned for the analysis of EMQ sub-scales. I invite the authors to report whether this was done. Furthermore, a justification for why participants with obesity were excluded from this analysis would be appreciated, as the association between BIS11 and HFS was significant with those participants included and would yield higher statistical power because of the higher amount of data points.

• line 339: Please elaborate: Why do the authors conclude that the size of the direct effect was reduced when including EMQ into the model when b, β, and t are larger for the model including EMQ?

• line 445: can the authors explain how controlling for hunger and sex is different to checking their possible effects. Was this done with separate models? Usually, the effect of a confounding covariate on the dependent variable can be inferred from the model that controls for said covariate. Is the approach described by the authors different to the approach used in experiment 1?

• line 495: the authors seem to have incorrectly interpreted the outcome of their mediation analysis. If both, the effect of X (HFS) on M (memory) and M (memory) on Y (impulsivity) are significant, and the direct effect of X on Y is insignificant, full mediation of the effect of X on Y through M exists. This finding would support the authors’ assumption of the vicious cycle model.

• Did the authors run the second mediation model – X (impulsivity) -> M (HFS) -> Y (memory) also for experiment 2 and 3 or do they draw their conclusion in favor of the trait model only from the absence of evidence for the vicious cycle model?

• line 639: the authors report that sex predicted EMQ scores but mention in the following sentence that TFEQD was related to total DFS score and EMQ remained a significant predictor. Did the authors run two different models, once with DFS and once with EMQ as dependent variable? This can be made clear to the reader by specifying all statistical models before reporting their outcomes.

• The manuscript does not mention a mediation analysis including the RCF ratio. Can the authors explain their reasoning for not performing this analysis? In general, the manuscript would provide more clarity if the conditions for conducting a mediation analysis were stated.

The manuscript is written in standard English and comprehensible for the reader. Especially the introduction presents the relevant background information and current state of the field as well as the research objective in a concise and logical way. The presentation of methods, specifically the statistical analyses, and results can be improved by providing a more consistent structure (see comments above).

• I suggest that the authors mention in the title that the association between HFS and impulsivity and memory has been found it humans, as this more precisely represents the content as well as the significance of the study.

• To be more precise and closer to the findings the authors report, I suggest speaking of high fat and sugar diet instead of unhealthy diet. An unhealthy diet can be more than increased intake of fat and sugar, for example increased salt or insufficient macro- or micronutrient intake. The fact that the authors revised their short title from “Unhealthy diet, memory and impulsivity” to “High fat and sugar diet, memory and impulsivity” indicates that they have made similar considerations already.

• For clarity, the authors could mention in the introduction that the presented manuscript comprises of three different studies, each building up on the previous.

• I invite the authors to use people-first language throughout the manuscript (e.g. line 290: “participants with obesity” instead of “obese participants”).

• Please describe the direction of the effect as well as the statistical (non-)significance in all figure legends so that readers do not have to go back to the main text but can grasp the main findings immediately.

• I recommend that the authors remove tables 2 and 3 and present the statistics in the text only. The reason for this is that it might be a bit surprising to see results of experiments 2 and 3 already in the section of experiment 1, but even more crucial that the reader must scroll back through a huge part of the manuscript to retrieve the information mentioned in the main text. If the authors want to keep tables 2 and 3 the statistical results should nevertheless be mentioned in the text.

• Linking statistical results to the proposed models regarding the inter-relationship between diet, impulsivity, and memory might be easier for the reader if the authors would consistently refer to the two models by their names, vicious cycle model and trait model, instead of referring to the figures 1a and 1b.

Reviewer #2: The authors respond to the existing findings that higher consumption of HFS foods may be linked to specific memory impairments, and that it may be related to higher impulsivity, by setting out to investigate whether / how the two effects may be related.

In the introduction, the authors propose to contrast predictions from the impulsivity hypothesis (they call it trait theory) and vicious cycle model (of obesity), which they outline and schematically present in Figure 1.

However, this figure does not correspond to what they write: For 1A, they propose that, “trait impulsivity is associated with higher HFS intake, which in turn leads to impaired memory” but their drawing shows a circular relation for FHS diet and memory, rather than a linear one. If the relation was intended to be circular, a justification should be presented. However, in the hypothesis, and overall discussion, they repeat the linear relation (impulsivity -> HFS -> memory).

In all cases, labelling is not ideal: We have impulsivity (for which a range of values can be measured by various tools), HFS intake (a range of values measured by self-report), and then IMPAIRED memory (some people have minimally impaired memory? This does not make sense. It should be ‘memory’).

Experiment 1 is well described, analyses are appropriate, figures helpful, and conclusions reasonable (tentative support for 1B model). A sound rationale is presented for running Experiment 2 with a memory task that does not rely on self-report; using a DDT as an additional measure of impulsivity; measuring hunger and BMI directly; not using a female-only sample.

A general point: DFS provided estimates of saturated fat and free sugar consumption rather than of total fats and sugar in the diet (olive oil and unsweetened fruit / juice are not counted for example) so it is related to participants’ ultra-processed food consumption (the authors mention WD in general discussion) – this should be mentioned.

Experiment 2 is well described, analyses appropriate, figures helpful, and conclusions reasonable (more in line with 1A model). Rationale for the final experiment was to include a memory task with an inbuilt inhibitory component as a test of 1B model, which presents a shift from the previous analyses that used memory and impulsivity as separate theoretical constructs. A directional hypothesis here is not warranted (but this is neither here or there as no relationship was found between the task performance and HFS consumption in E3).

Why did authors use a money-based rather than food-based discounting task?

Please explain how height and weight were recorded by the experimenter(s).

The objective of not testing a female-only sample was only partially realised; the sample was predominantly female both here and in Experiment 3. This needs to be acknowledged, especially because there were some sex differences noted.

Was the sample in E2 overlapping with previous study, drawn from the same pool?

Experiment 3 is likewise well designed and conducted, with good reporting, analyses, figures, and conclusions. In discussion authors refer to findings about sugar and fat consumption and Tables 8 and 9 – but there are no tables included beyond 6. What else may be missing? What about other experiments – was it the fat score of the HFS that carried the effects described? This deserves some consideration, not least because title refers to habitual intake of fat and sugar.

General discussion is reasonable, except here perhaps it should be stated more explicitly (conclusion?) that a strong possibility exists that neither of the two simple 1A and 1B models could, in principle, account for the complex relationships investigated in the studies, or the results that they present.

The authors are appropriately cautious when they describe their results as associations, not implying that directionality could be established, except in line 752-753. This should be rephrased. Obviously, it is fine to speculate about directionality when examining various existing (mostly non-human) manipulative experimental findings and discussing theoretical models.

The models of impulsivity, memory impairment, and their links with unhealthy HFS diet were proposed as accounts for obesity. In general discussion, authors state that BMI status of participants was not related to study variables (impulsivity, memory) – what about their HFS scores? They speculate that in long term those favouring HFS diet may become obese, but weight status at the age which they recruited is strongly predictive of later weight.

In most participants, BMI scores seem to have been lower than population average (and ranges show scores of BMI as low as 16); what about their HFS scores, memory performance, impulsivity? It would be good to know because if the sample were eating better than average diets, the effects that the experimenters were looking for may have been hard to detect.

Line 250: repression should be regression.

Abstract: ‘also’ twice in the last sentence; please replace ‘further support’ with ‘some support’ and look closely at this conclusion – what do you mean by ‘trait impulsivity being a risk factor for poor dietary choice which is then exacerbated by diet?’ – impulsivity being in a vicious circle with diet irrespective of memory? A third model?

Overall, this is a well-written paper that presents findings which complement the existing literature. I would recommend it for publication after the queries have been answered.

6. PLOS authors have the option to publish the peer review history of their article (what does this mean?). If published, this will include your full peer review and any attached files.

Reviewer #1: No

Reviewer #2: No

---

## [Author Response · Author response to Decision Letter 0]

29 Jun 2023

Responses to reviewers comments.

We would first like to thank the two reviewers for their helpful comments. We detail responses to those comments below (in italics for clarity in the attached Word file) and have marked the revised text in blue font in the revised track-changed version of the resubmitted MS.

Both reviewers noted some difficulty understanding and interpreting the outcomes of the moderation analyses we conducted, and inconsistency in how we reported this across the three experiments. On re-reading the submitted MS we agreed with their views, and have made more substantial changes to the sections detailing the mediation analyses than they requested, especially for Experiments 2 and 3, to give a more accurate account of these findings. In particular, we had not properly reported outcomes of mediation for both vicious cycle and trait models for those experiments. Because of the multiple models tested, we have now summarised the key outputs from mediation in new tables to allow the reader a more visual way of seeing how consistent these were across the multiple measures in each experiment. Further details are described in our responses to reviewers below.

Reviewer #1:

The manuscript is technically sound, the experiments were conducted rigorously, and the sample sizes are sufficient and justified by power analyses. Most of the conclusions drawn from the data are appropriate and do not overreach. Nevertheless, some specifications would improve the understanding of how the data was acquired and conclusions drawn.

We thank the reviewer for these positive comments.

• line 238: How much time passed between the assessment of the TFEQ data and DFS, EMQ, BIS11, and DI data? How could this delay affect the association between the two data sets?

There was a maximum of 2 months between the TFEQ and main online study, with an average gap of 3 weeks. We have noted this in the revised procedure.

• line 245: please provide or reference the standard formula for BMI

We have now added the classic 1972 paper which is usually attributed with the wider use of BMI as a measure. We’ve added this when we first introduce BMI (line XXX) of the revised MS.

• line 250: the authors mistyped “regression”

Thanks for spotting this: a Word autocorrection we should have spotted.

• line 319: What are the “different dietary measures” the BIS11 sub-scales were related to?

Corrected to “DFS total score”

• line 320: I invite the authors to precisely report the statistics for the association of BIS sub-scales with DFS and refrain from generalized statements (“to some extend related”), especially when the association of BISmot with DFS did not reach significance.

We have revised the wording to be more precise, although we also note that the correlation between DFS and BISmot (0.20) in E1 may not have been significant because of lack of power, so do not draw strong conclusions from that.

• line 329: The authors mistyped the alpha level (p < 0.10 instead of p < 0.01)

Corrected

• line 400: can the authors explain why they deemed controlling for hunger at the time of testing necessary? Is there any literature linking hunger state to their measures of impulsivity or memory? 

We controlled for hunger because we have recently developed a model of human hunger that implicates the hippocampus, and wanted to ensure that any effects of memory were not secondary to state differences in hunger at the time of testing. We have added a sentence to the discussion after Experiment 1 to explain this briefly.

Were all memory and impulsivity measures controlled for hunger? The manuscript only mentions BIS11 (line 483) but not DDT (line 480) or EMQ (line 462)

We apologise for the under-reporting of models for Experiment 2: we tried to be concise and not repeat information from Experiment 1 but in this case that lacked clarity. We have now revised to fully specify the regression models.

• I invite the authors to address in the discussion how they interpret the finding that BISatt was most consistently associated with HFS and involved in the mediation analysis to test the two possible models? Can they speculate why this sub-component of impulsivity, in contrast to the others, is involved in HFS intake and hippocampus-related memory?

We now discuss why the attention measure was the most consistently associated in the general discussion.

• line 603: When did the taste test in experiment 3 happen and how much later were hunger rating assessed?

Sorry for that lack of clarity: we have added more procedural detail. The test session started with a hunger rating, followed by the taste test (not relevant to these analyses) and the key memory tests started roughly 10 minutes after the hunger rating: this is now clear in the revised text.

• line 616: Could the authors explain why a higher ratio of total unrehearsed to rehearsed items recalled can be regarded as a measure of memory inhibition?

In the RCF task, the classic result is that rehearsal of some items from a category enhances recall of those items, but inhibits recall of the unrehearsed items (as detailed in the original RCF paper by Anderson and colleagues): the ratio measure captures that effect, with lower scores equating to greater inhibition. We have added that reference to the description in data analysis.

• line 696: the authors mention evidence that habitual intake of fat, alone or in combination with sugar, underlies the association of DFS scores with memory and impulsivity. Which analyses provide this evidence? Please mention in the statistical analysis section that models with the DFS sub-scores were conducted and report the statistical findings in the results section. Were the sub-scores also used for the mediation analyses? If not, the authors should consider these analyses. And were these analyses done only for experiment 3? Furthermore, the text mentions tables 8 and 9, which are not included in the manuscript.

Our apologies: in an earlier draft of this MS we included analysis of the relationship between the sub-scales of the DFS and memory/impulsivity scores. However, we concluded that because the numbers of items in each nutritional sub-group (fat, fat/sugar and sugar) differ markedly, those analyses were not reliable. The errant paragraph the reviewer refers to should have been removed and now has been.

• line 730: the authors state that to their knowledge the present manuscript is the second to examine how habitual intake of fat and sugar relate to impulsivity. In fact, at least three further studies investigated this relationship: 1) Steele et al., Appetite, 2021 (DOI: 10.1016/J.APPET.2021.105292) report no association of fat but sugar with impulsive choices, 2) Hartmann et al., J Neuroendocrinology, 2020 (DOI: 10.1111/JNE.12917), found no association of two extreme groups of low and high consumers of fat and sugar diet with the UPPS, 3) Hartmann et al., Appetite, 2023 (DOI: 10.1016/J.APPET.2023.106477) found no association of two extreme groups of low and high consumers of fat and sugar diet with the BIS15

We thank the reviewer for drawing our attention to these papers. We read the 2023 Hartmann paper with great interest when it was first published, and had anticipating discussing those findings in the revision of this MS. But we could not have included it in the original submission as our MS was submitted in March, prior to publication of that paper. We agree that we should, however, have discussed the other two, although notably both had quite small sample sizes so how reliable the lack of difference the report was is unclear. All 3 papers are now discussed in this revision.

• line 737: the authors mention the correlation of BIS11 with the Dickman dysfunctional scale. Please mention this test in the statistics and results section

Sorry for that omission: we have added a note on the tests we did in the data analysis section of Experiment 1, a very brief comment on those inter-correlations in the results of Experiment 1, and now include the full correlation matrix as a Supplementary Table. When protected for multiple comparisons, only the correlation between BISatt and DIdys was significant (although for both impulsivity measures their sub-scales were inter-correlated).

• line 742: the authors state that inhibitory failure is a key feature of the vicious cycle model. Could the authors discuss why the RCF task ratio, which is supposed to measure inhibitory control, is not associated with HFS in contrast to other measures of memory?

We now discuss the implications of the lack of evidence with the RCF inhibitory measure in E3.

• line 781: Hartmann et al., Appetite, 2023 (DOI: 10.1016/J.APPET.2023.106477) investigated the association of habitual fat and sugar intake with striatal and prefrontal function using a working memory task. They did not find any association of HFS with behavioral or neural measures of PFC function.

This study was very timely! We have now noted that finding.

• Figure 1a shows a feedback loop between HFS and memory, but this feedback is never mentioned in the text introducing the trait model. I strongly recommend removing this feedback loop and only show a linear relation with effects of HFS on memory in line with the background provided in the text.

We agree and have amended Figure 1a accordingly, and agree that fits better with the text and the trait model concept. 

• In various occasions throughout the manuscript the authors mention moderation analysis although I assume they meant mediation.

Thank you for spotting this: we found 3 instances of that error and have correct them all.

• The causal relation between impulsivity, HFS, and memory for the trait model is not always presented in a coherent way. In line 131 the authors write that trait impulsivity is associated with higher HFS intake, which in turn leads to impaired memory. But in line 146 the authors state that the trait model does not make any suggestions about any negative consequences of HFS intake. In the same sentence the authors claim again that HFS might influence memory, but it would be independent from impulsivity. I invite the authors to explicitly state the relationship of factors in the trait model and converge this on to the mediation analyses, describing how significance in which pathway is in favor of the trait model.

We agree this was unclear. We have revised this to make clear that the primary prediction of the trait model is that trait impulsivity is associated with higher DFS. We then clarify (after introducing the VCM mediation test) that a secondary prediction of the trait model is that higher DFS encouraged by trait impulsivity) then leads to impaired memory. We hope that is now clearer.

• In my opinion, the authors should be more careful with the conclusion they can draw from their data about the vicious cycle model and mention some limitations of the experimental design. The vicious cycle model proposed by Hargrave, Jones & Davidson (2017) and Davidson et al. (2019) mentions food cue reactivity as the main link between impaired memory inhibition, and though impulsivity and food cue reactivity can be related, van den Akker et al. (2014) recommends a multimodal assessment of both constructs. Furthermore, the vicious cycle model relies on food-related memories and memories of post-ingestive rewards. The measures of memory presented in this manuscript pertain to everyday memory (EMQ), spatial memory (4MT), and short-term memory (RCF task). I invite the authors to consider these limitations and discuss if and how the presented measures relate to food cue reactivity and food memory and what conclusions can be drawn in favor of the vicious cycle model.

The vicious circle model (VCM: e.g., Hargraves, Jones & Davidson, 2017 – figure 1) specifies the following chain of events: excess intake of a diet high in fat and/or sugar causes hippocampal dysfunction, which causes impaired memory inhibition, which then leads to increased food cue reactivity. The reviewer raises two points about this process relevant to our approach to testing it.

First, we measured hippocampal dysfunction using the 4MT and EMQ. The RCF task measures memory inhibition – and so relatedly hippocampal function. Crucially the VCM does not specify that the hippocampal dysfunction caused by a Western style diet is specific to food/food memory and so the impairment can be measured using the type of general measures we employed. We now note in the introduction (LN137-155) that these particular constructs of the VCM can be measured using food-non-specific tasks (i.e. standard neuropsych measures) because the impairment to the hippocampus is general.

The second issue concerns the measurement of food cue reactivity. Our perspective on this is shaped by the impact that failure of memory inhibition has. When sated, food cues that would not normally retrieve an excitatory memory due to memory inhibition, do so when there is hippocampal dysfunction. Thus, a failure of memory inhibition results in the retrieved memory being more excitatory. As this is a consequence of hippocampal dysfunction and hence impaired memory inhibition, these serve as a proxy measure of greater excitatory food memory retrieval when sated. We have now made this clearer in the Introduction (LN137-155)

Impulsivity makes a special and unique contribution to how detrimental this memory inhibition failure is. High levels of impulsivity will make it much harder to resist experiencing a desirable food memory (e.g., remembering how delicious a mars bar is). Hence, impulsivity is not serving as a measure of food cue reactivity. Rather degree of impulsivity indicates whether or not the more desirable memory will be acted upon.

Overall, the authors chose the appropriate statistical models to answer their research questions. The statistical analysis has been conducted rigorously, though it is not always clear to the reader how exactly the models were specified and what is the entirety of models that were run by the authors. I advise the authors to provide a section explaining all the models that were run in such a form that it becomes clear which variables were used as independent variable, dependent variable, and covariate (dv ~ iv + covar). This can be shown in a general manner and then specified for the respective variables for each experiment. 

We have revised all the Data analysis sections, making clearer the different steps in each regression analyses, with the overall approach detailed in the analysis section of Experiment 1, and then for brevity specific details only for the other two Experiments.

Furthermore, it would be beneficial to showcase the mediation models in the conventional graph, depicting the predictor, mediator, and outcome variable as well as direct and indirect effects. The same figures should be used to explain the results of the mediation analyses, showing a and b for the two paths of the indirect effect, the indirect effect ab, the direct effect c’, and the total effect c.

On re-reading the MS we also felt our reporting of the mediation analyses needed to be clearer, especially for Experiments 2 and 3 where our original desire to be concise led to significant under-reporting of outcomes, with mediation for Trait models minimally reported. The results sections for mediation for both Experiments 2 and 3 have been substantially revised as a consequence. We now include new Figures showing key mediation models for the three experiments, although as we used multiple measures of memory and impulsivity we also summarise all models in additional tables as well as providing more detailed summaries in Supplementary tables. Readers will also be able to access the full analysis outputs and the data through the linked data repository on publication.

• How did the authors define statistical outliers in their data? 

Outliers in all three Experiments were checked initially using visual inspection of Box and/or Q-Q plot. Where there were potential outliers, these were formally checked by analysing z-scored data, defining those which fell outside the 95% confidence interval as outliers.

Were all participants with BMI > 30 statistical outliers or excluded based on their body weight status obese?

Analysis of z-scores identified three of the four participants with BMI >30 as outliers (those with BMI of 48.49, 40.96 and 31.22). The fourth person had a BMI of 31.11, and it seemed illogical to keep that person in the additional analyses excluding outliers. We have clarified this in the revised MS.

• line 264: The authors seem to have misspecified the mediation model to test the vicious cycle model (“any effect of memory on DFS would be mediated by impulsivity”). In line 334 they specify the model to test the vicious cycle model as “whether EMQ mediated the relationship between DFS and BIS11”

Line 264 was absolutely wrong. We have corrected and apologise.

• line 279: For other researchers being able to replicate the authors’ analyses I recommend describing how the robust models were different to the conventional models.

We have detailed the method we used to conduct the Robust model in the revised data analysis section.

• line 253: Following up on my general notion that the manuscript would benefit from a clear description of the statistical models, I invite the authors to explain whether in step 3 TFEQ was added to the model from step 1 or step 2.

This is now clear in the revised data analysis. In brief, if none of the covariates added factors in Step 2 were significant, the TFEQ factors were added to the base model (Step 1) but if any of the factors in Step 2 had a significant effect they were retained in Step 3.

• line 319: participants with obesity were excluded from the analysis of the BIS11 sub-scales but a similar procedure is not mentioned for the analysis of EMQ sub-scales. I invite the authors to report whether this was done. Furthermore, a justification for why participants with obesity were excluded from this analysis would be appreciated, as the association between BIS11 and HFS was significant with those participants included and would yield higher statistical power because of the higher amount of data points.

We apologise: as with memory, as the initial analysis found no effects of age or BMI, we ran these follow-up analyses with the full dataset to maximise power. The test has been corrected: the reported data were correct.

• line 339: Please elaborate: Why do the authors conclude that the size of the direct effect was reduced when including EMQ into the model when b, β, and t are larger for the model including EMQ?

Our error: “reduced” should have read “increased”. How that happened we don’t know, but thanks for spotting and its now corrected.

• line 445: can the authors explain how controlling for hunger and sex is different to checking their possible effects. Was this done with separate models? Usually, the effect of a confounding covariate on the dependent variable can be inferred from the model that controls for said covariate. Is the approach described by the authors different to the approach used in experiment 1?

It was the same approach as Experiment 1, and we were trying to be concise. We have rephrased to make this clearer.

• line 495: the authors seem to have incorrectly interpreted the outcome of their mediation analysis. If both, the effect of X (HFS) on M (memory) and M (memory) on Y (impulsivity) are significant, and the direct effect of X on Y is insignificant, full mediation of the effect of X on Y through M exists. This finding would support the authors’ assumption of the vicious cycle model.

This section has been substantially rewritten to give a fuller account of these analyses, and to make clearer the separate models for VCM and trait accounts.

• Did the authors run the second mediation model – X (impulsivity) -> M (HFS) -> Y (memory) also for experiment 2 and 3 or do they draw their conclusion in favor of the trait model only from the absence of evidence for the vicious cycle model?

We did run both mediation models but our initial reporting did not make this clear enough: the substantial revisions here hopefully address the concerns and give a concise summary of these models.

• line 639: the authors report that sex predicted EMQ scores but mention in the following sentence that TFEQD was related to total DFS score and EMQ remained a significant predictor. Did the authors run two different models, once with DFS and once with EMQ as dependent variable? This can be made clear to the reader by specifying all statistical models before reporting their outcomes.

The models were the same as the two previous experiments, with the addition of sex (added at Step 2) and that has been clarified in the revised data analysis section.

• The manuscript does not mention a mediation analysis including the RCF ratio. Can the authors explain their reasoning for not performing this analysis? In general, the manuscript would provide more clarity if the conditions for conducting a mediation analysis were stated.

Mediation models aim to interpret how different correlated relationships interact: therefore, it only makes sense to run mediation where there was an effect, and for the RCF ratio this was not the case. We now make that clear in the revised MS. 

The manuscript is written in standard English and comprehensible for the reader. Especially the introduction presents the relevant background information and current state of the field as well as the research objective in a concise and logical way. The presentation of methods, specifically the statistical analyses, and results can be improved by providing a more consistent structure (see comments above).

• I suggest that the authors mention in the title that the association between HFS and impulsivity and memory has been found it humans, as this more precisely represents the content as well as the significance of the study.

We have made clear in the title this was in humans.

• To be more precise and closer to the findings the authors report, I suggest speaking of high fat and sugar diet instead of unhealthy diet. An unhealthy diet can be more than increased intake of fat and sugar, for example increased salt or insufficient macro- or micronutrient intake. The fact that the authors revised their short title from “Unhealthy diet, memory and impulsivity” to “High fat and sugar diet, memory and impulsivity” indicates that they have made similar considerations already.

We have edited to remove mention of unhealthy when referring to the diet

• For clarity, the authors could mention in the introduction that the presented manuscript comprises of three different studies, each building up on the previous.

We have added a sentence at the end of the introduction to confirm this was our approach.

• I invite the authors to use people-first language throughout the manuscript (e.g. line 290: “participants with obesity” instead of “obese participants”).

We have rephrased all instances where we had previously used obese participants accordingly.

• Please describe the direction of the effect as well as the statistical (non-)significance in all figure legends so that readers do not have to go back to the main text but can grasp the main findings immediately.

We note the reviewers point but felt on reflection that the addition of this information to the Figure legends would make them unwieldy and potentially confusing. Hence we wish to leave the legends as they currently are.

• I recommend that the authors remove tables 2 and 3 and present the statistics in the text only. The reason for this is that it might be a bit surprising to see results of experiments 2 and 3 already in the section of experiment 1, but even more crucial that the reader must scroll back through a huge part of the manuscript to retrieve the information mentioned in the main text. If the authors want to keep tables 2 and 3 the statistical results should nevertheless be mentioned in the text.

Although we agree that having to scroll back to these Tables is cumbersome, the Tables themselves give a concise summary of the replicability of the findings across studies. So we have taken the compromise of summarising the statistics in each section, as suggested, but retaining the two Tables as useful overall summaries of the outcomes.

• Linking statistical results to the proposed models regarding the inter-relationship between diet, impulsivity, and memory might be easier for the reader if the authors would consistently refer to the two models by their names, vicious cycle model and trait model, instead of referring to the figures 1a and 1b.

We now always use the name, vicious cycle or trait model, when cross-referring to figure 1.

Reviewer #2: The authors respond to the existing findings that higher consumption of HFS foods may be linked to specific memory impairments, and that it may be related to higher impulsivity, by setting out to investigate whether / how the two effects may be related.

In the introduction, the authors propose to contrast predictions from the impulsivity hypothesis (they call it trait theory) and vicious cycle model (of obesity), which they outline and schematically present in Figure 1.

However, this figure does not correspond to what they write: For 1A, they propose that, “trait impulsivity is associated with higher HFS intake, which in turn leads to impaired memory” but their drawing shows a circular relation for FHS diet and memory, rather than a linear one. If the relation was intended to be circular, a justification should be presented. However, in the hypothesis, and overall discussion, they repeat the linear relation (impulsivity -> HFS -> memory).

We have amended the figure to show a linear relation (see also response to Reviewer 1)

In all cases, labelling is not ideal: We have impulsivity (for which a range of values can be measured by various tools), HFS intake (a range of values measured by self-report), and then IMPAIRED memory (some people have minimally impaired memory? This does not make sense. It should be ‘memory’).

We have changed the label impaired memory to memory, and quite agree that was misleading.

Experiment 1 is well described, analyses are appropriate, figures helpful, and conclusions reasonable (tentative support for 1B model). A sound rationale is presented for running Experiment 2 with a memory task that does not rely on self-report; using a DDT as an additional measure of impulsivity; measuring hunger and BMI directly; not using a female-only sample.

Thank you for these positive comments.

A general point: DFS provided estimates of saturated fat and free sugar consumption rather than of total fats and sugar in the diet (olive oil and unsweetened fruit / juice are not counted for example) so it is related to participants’ ultra-processed food consumption (the authors mention WD in general discussion) – this should be mentioned.

We have added a note to limitations.

Experiment 2 is well described, analyses appropriate, figures helpful, and conclusions reasonable (more in line with 1A model). Rationale for the final experiment was to include a memory task with an inbuilt inhibitory component as a test of 1B model, which presents a shift from the previous analyses that used memory and impulsivity as separate theoretical constructs. A directional hypothesis here is not warranted (but this is neither here or there as no relationship was found between the task performance and HFS consumption in E3).

Why did authors use a money-based rather than food-based discounting task?

At the time E2 was designed we were focussed on general impulsivity, but with hindsight recognise that a food-related measure would have been more appropriate. We now discuss this under limitations and future directions.

Please explain how height and weight were recorded by the experimenter(s).

We have added details of the balance and stadiometer used

The objective of not testing a female-only sample was only partially realised; the sample was predominantly female both here and in Experiment 3. This needs to be acknowledged, especially because there were some sex differences noted.

We now note in limitations.

Was the sample in E2 overlapping with previous study, drawn from the same pool?

All three samples were drawn from the same study population, but collected in different years.

Experiment 3 is likewise well designed and conducted, with good reporting, analyses, figures, and conclusions. In discussion authors refer to findings about sugar and fat consumption and Tables 8 and 9 – but there are no tables included beyond 6. What else may be missing? 

Our apologies: that discussion paragraph should have been removed (an earlier draft included analyses by diet sub-component but we concluded those analyses were not reliable: see earlier comments to reviewer 1.)

What about other experiments – was it the fat score of the HFS that carried the effects described? This deserves some consideration, not least because title refers to habitual intake of fat and sugar.

We agree that this could be interesting, but conclude that the DFS is not the right tool to allow a detailed analysis by dietary component since it has multiple items for some components (fat, fat+sugar) but few for sugar. Therefore lack of significance with sugar might just reflect restricted variance.

General discussion is reasonable, except here perhaps it should be stated more explicitly (conclusion?) that a strong possibility exists that neither of the two simple 1A and 1B models could, in principle, account for the complex relationships investigated in the studies, or the results that they present.

We hope that the revised discussion now makes this clearer.

The authors are appropriately cautious when they describe their results as associations, not implying that directionality could be established, except in line 752-753. This should be rephrased. Obviously, it is fine to speculate about directionality when examining various existing (mostly non-human) manipulative experimental findings and discussing theoretical models.

In this case we were explicitly discussing the mediation analysis for the VCM model, which had a directional hypothesis, so believe our wording is acceptable in that context.

The models of impulsivity, memory impairment, and their links with unhealthy HFS diet were proposed as accounts for obesity. In general discussion, authors state that BMI status of participants was not related to study variables (impulsivity, memory) – what about their HFS scores? They speculate that in long term those favouring HFS diet may become obese, but weight status at the age which they recruited is strongly predictive of later weight.

This is an interesting point, however we felt including this would have the cost of both further lengthening the MS and not addressing the specific research questions we originally set out to answer. For these reasons we chose not to add this extra test.

In most participants, BMI scores seem to have been lower than population average (and ranges show scores of BMI as low as 16); what about their HFS scores, memory performance, impulsivity? It would be good to know because if the sample were eating better than average diets, the effects that the experimenters were looking for may have been hard to detect.

Unfortunately there is not normative data on the DFS so it is not currently possible to address this point.

Line 250: repression should be regression.

Corrected

Abstract: ‘also’ twice in the last sentence; please replace ‘further support’ with ‘some support’ and look closely at this conclusion – what do you mean by ‘trait impulsivity being a risk factor for poor dietary choice which is then exacerbated by diet?’ – impulsivity being in a vicious circle with diet irrespective of memory? A third model?

We made the two minor wording changes and deleted the final clause for brevity.

Overall, this is a well-written paper that presents findings which complement the existing literature. I would recommend it for publication after the queries have been answered.

---

## [Decision Letter · Decision Letter 1]

7 Aug 2023

Habitual intake of fat and sugar is associated with poorer memory and greater impulsivity in humans.

PONE-D-23-04749R1

Dear Dr. Yeomans,

We’re pleased to inform you that your manuscript has been judged scientifically suitable for publication and will be formally accepted for publication once it meets all outstanding technical requirements.

Kind regards,

Hans-Peter Kubis, PD. Dr. rer. nat.

Academic Editor

PLOS ONE

Additional Editor Comments (optional):

Reviewers' comments:

Reviewer's Responses to Questions

**Comments to the Author**

1. If the authors have adequately addressed your comments raised in a previous round of review and you feel that this manuscript is now acceptable for publication, you may indicate that here to bypass the “Comments to the Author” section, enter your conflict of interest statement in the “Confidential to Editor” section, and submit your "Accept" recommendation.

Reviewer #1: All comments have been addressed

Reviewer #2: All comments have been addressed

2. Is the manuscript technically sound, and do the data support the conclusions?

Reviewer #1: Yes

Reviewer #2: Yes

3. Has the statistical analysis been performed appropriately and rigorously? 

Reviewer #1: Yes

Reviewer #2: Yes

4. Have the authors made all data underlying the findings in their manuscript fully available?

Reviewer #1: Yes

Reviewer #2: Yes

5. Is the manuscript presented in an intelligible fashion and written in standard English?

Reviewer #1: Yes

Reviewer #2: Yes

6. Review Comments to the Author

Reviewer #1: (No Response)

Reviewer #2: Thank you for responding to comments. This is a good and interesting article. I have no further requests.

7. PLOS authors have the option to publish the peer review history of their article (what does this mean?). If published, this will include your full peer review and any attached files.

Reviewer #1: No

Reviewer #2: No

---

## [Editor Report · Acceptance letter]

16 Aug 2023

PONE-D-23-04749R1 

Habitual intake of fat and sugar is associated with poorer memory and greater impulsivity in humans. 

Dear Dr. Yeomans:

I'm pleased to inform you that your manuscript has been deemed suitable for publication in PLOS ONE. Congratulations! Your manuscript is now with our production department. 

Kind regards, 

on behalf of

Dr. Hans-Peter Kubis 

Academic Editor

PLOS ONE